# Autonomous Functional Play with Correspondence-Driven Trajectory Warping

**William Liang**[1,2]   **Sam Wang**[1]   **Hung-Ju Wang**[1,3]
**Osbert Bastani**[1]   **Yecheng Jason Ma**[1,3†]   **Dinesh Jayaraman**[1†]
[1]University of Pennsylvania   [2]University of California, Berkeley   [3]Dyna Robotics

https://tether-research.github.io

## Abstract

The ability to conduct and learn from interaction and experience is a central challenge in robotics, offering a scalable alternative to labor-intensive human demonstrations. However, realizing such "play" requires (1) a policy robust to diverse, potentially out-of-distribution environment states, and (2) a procedure that continuously produces useful robot experience. To address these challenges, we introduce Tether, a method for autonomous functional play involving structured, task-directed interactions. First, we design a novel open-loop policy that warps actions from a small set of source demonstrations ($\leq 10$) by anchoring them to semantic keypoint correspondences in the target scene. We show that this design is extremely data-efficient and robust even under significant spatial and semantic variations. Second, we deploy this policy for autonomous functional play in the real world via a continuous cycle of task selection, execution, evaluation, and improvement, guided by the visual understanding capabilities of vision-language models. This procedure generates diverse, high-quality datasets with minimal human intervention. In a household-like multi-object setup, our method is the first to perform many hours of autonomous multi-task play in the real world starting from only a handful of demonstrations. This produces a stream of data that consistently improves the performance of closed-loop imitation policies over time, ultimately yielding over 1000 expert-level trajectories and training policies competitive with those learned from human-collected demonstrations.

## 1 Introduction

Recent advances in robotic manipulation have been powered by imitation learning policies (Zhao et al., 2023; Chi et al., 2023; Zhao et al., 2024; Black et al., 2024; Kim et al., 2024; Brohan et al., 2022; 2023; Collaboration et al., 2024) trained on real-world teleoperated demonstrations. In all these cases, the human effort involved in teaching a skill is substantial: while there are continued efforts to simplify teleoperation interfaces (Zhao et al., 2023; Shafiullah et al., 2023; Chi et al., 2024; Cheng et al., 2024; Fu et al., 2024), such demo datasets can fundamentally only scale linearly with human time, and these data-hungry policy architectures need large spatially and semantically diverse datasets in order to generalize usefully (Lin et al., 2024). In this paper, we propose an alternative paradigm: inspired by *functional play* in developmental psychology (Piaget, 1962; Smilansky, 1968)—characterized by structured, task-directed interactions and repetitive practice—we design Tether, a method for autonomous robot play that circumvents the human effort bottleneck.

Our method involves two key components. First, autonomous functional play requires an extremely robust policy that can reliably recover from mistakes and out-of-distribution states. Without relying on massive datasets for training large neural policy architectures, we instead design a new open-loop policy class that specifically supports generalization with few demonstrations. Specifically, our architecture exploits the remarkable leaps in semantic image keypoint correspondences: given a new scene with potentially new task-relevant object instances in new spatial layout with new distractors, it first computes keypoint correspondences with the demo images, selects the closest-matched demo,

---

[†]Equal advising.

computes 3D transformations associated with each correspondence, and accordingly warps the robot trajectory to fit the new scene. Validated on 12 manipulation tasks in a household-like setting, we show that our policy surpasses the performance of alternative methods, including those that rely on foundation models or pretraining with large robotics or internet datasets.

Second, we run our policy within a cyclic functional play procedure that autonomously produces data for continuous downstream policy training. For a collection of tasks with few human demonstrations each, we query a vision-language model (VLM) to repeatedly plan and select tasks our policy should attempt. This is real-world play without any manual resets: the procedure naturally induces resets with a growing initial state distribution as the object configurations drift away from the beginning of play. Additionally, to evaluate the suboptimal play data, we query a VLM to detect successful executions, which are then used downstream for filtered imitation learning. We show that this procedure can collect over 1000 new expert-level demonstrations across 26 hours with minimal human intervention (5 cases requiring a minute total of human time, 0.26% of executions). We also validate that this stream of newly generated demonstrations progressively enhances the training of imitation learning policies, which consistently improve with more play data and reach high success rates competitive with counterparts trained on human-collected demonstrations.

In summary, our contributions are:

1. A keypoint correspondence-driven trajectory warping policy that exhibits impressive spatial and semantic robustness for diverse manipulation tasks.
2. A multi-task VLM-guided play procedure that generates diverse expert-level demonstrations over many hours, powering downstream neural policy training.

## 2 RELATED WORK

**Robustness in Imitation Learning.** To build robust policies that excel in diverse environments, many prior efforts turn to scaling data and policy architectures, often with foundation models or large human-collected demo datasets (Khazatsky et al., 2024; Collaboration et al., 2024; AgiBot-World-Contributors et al., 2025). Prominent techniques include training representations for robotics on non-robot data (e.g., human videos) (Nair et al., 2022; Ma et al., 2023; Shi et al., 2025), querying vision-language models (VLMs) or large language models (LLMs) (Nasiriany et al., 2024; Goetting et al., 2024; Fang et al., 2024), and finetuning vision-language-action models (VLAs) (Black et al., 2024; NVIDIA et al., 2025; Intelligence et al., 2025).

Another class of approaches design strong built-in priors for models trained on much fewer demos, with some executing actions open-loop. These works build action affordances or primitives (Kuang et al., 2024; Haldar & Pinto, 2025), retrieve from existing datasets (Du et al., 2023; Memmel et al., 2024; Xie et al., 2025), leverage pretrained representations and models (Pari et al., 2021; Burns et al., 2023; Shi et al., 2024), or exploit 3-D scene geometry (Rashid et al., 2023; Goyal et al., 2024; Ke et al., 2024; Ze et al., 2024). Closest to our work are methods that operate on visual semantic keypoint correspondences. Like object-centric approaches (Devin et al., 2018; Qian et al., 2024; Zhao et al., 2025), they benefit from recent advances in scene understanding and are naturally robust to distractors, yet provide higher spatial precision and avoid the rigid "objectness" assumptions that fail on deformable objects and granular particles. One class of approaches tracks keypoints through frames of human or robot videos and retargets the dense trajectory to the desired setting (Wen et al., 2023; Bharadhwaj et al., 2024; Ren et al., 2025). Another class instead uses keypoint correspondences as a compact trajectory representation. Among these, KAT (Di Palo & Johns, 2024) queries an LLM with keypoints to generate open-loop actions, while P3-PO (Levy et al., 2024) and SKIL (Wang et al., 2025) input keypoints to point-conditioned policies. We too use keypoint correspondences, but demonstrate the advantages of a more direct approach: to select and warp a demonstrated trajectory to fit the new scene. We compare against KAT (Di Palo & Johns, 2024) in our experiments.

**Autonomous Data Generation.** To reduce the need for large human-collected manipulation datasets, recent efforts have explored data generation for policy learning. Some works collect data in simulation by querying foundation models to propose and solve tasks (Ha et al., 2023; Wang et al., 2024) or leveraging privileged simulation state to adapt demos for new scene configurations (Mandlekar et al., 2023; Jiang et al., 2025; Lin et al., 2025a). However, simulation-based approaches struggle with sim-to-real transfer within cluttered, unstructured environments, which remains an open challenge

especially for tasks involving complex contacts and vision-based policies trained on synthetic renders (Blanco-Mulero et al., 2024; Yu et al., 2024; Lin et al., 2025b). Alternatively, other works generate data directly in the real world. Some require policies trained on hundreds of human-collected demos (Zhou et al., 2024; Mirchandani et al., 2024), which still pose a bottleneck when transferring to new tasks and environments. Most similar to our work are methods that initialize from few demos using specially-designed policies. Manipulate-Anything (Duan et al., 2024b) introduces a zero-shot foundation model policy for autonomous data collection, but performing multiple rounds of foundation model inference significantly hinders throughput, culminating in under 50 demos; additionally, environment reset is not considered. In contrast, we introduce a system that autonomously produces over 1000 demonstrations in the real world with minimal human intervention.

## 3 ROBUST IMITATION AND AUTONOMOUS PLAY

We formulate robot manipulation as a Controlled Markov Process (CMP). A CMP is represented as a tuple $M = (S, A, \mathcal{T})$, where $S$ is the set of states, $A$ is the set of actions, $\mathcal{T} : S \times A \to \Delta(S)$ is the transition dynamics function.

In the imitation learning problem, for a given CMP $M$ and a desired task, we have a dataset of $N$ expert demonstrations $\mathcal{D} = \{\tau_1, \tau_2 \ldots, \tau_N\}$ that perform that task, with trajectories $\tau_i = \{s_1, a_1, s_2, a_2, \ldots\}$. In practice, our robot does not have direct access to state $s_t$ and instead receives visual observations $o_t = O(s_t)$, consisting of two third-person RGB camera views from the left and right sides, as shown in Figure 1. We assume no significant occlusions or partial observability. The actions $a_t$ are the 6-DOF pose of robot gripper and the gripper's binary open/close command. Given this demonstration dataset $\mathcal{D}$, our goal is to learn a policy $\pi : O \to \Delta(A)$ for the task.

As motivated in the introduction, today's standard imitation learning approaches require extensive human effort that is difficult to scale. Our paper directly addresses this problem. First, to produce effective behaviors with minimal human demo collection, we introduce an open-loop trajectory warping-based policy that generates action plans from a few training demonstrations (Section 3.1), much fewer than standard neural policies. Next, this policy serves as a robust bootstrap for autonomous data generation, powering a VLM-guided functional play procedure (Section 3.2) that produces demos for the downstream training of more expressive and powerful policy architectures.

### 3.1 TRAJECTORY WARPING WITH KEYPOINT CORRESPONDENCES

Towards spatially and semantically robust imitation against large variations in the initial state, Tether leverages semantic visual priors in the form of image correspondence matching algorithms to interpolate within and generalize beyond a few demonstrations.

Our policy is designed to run open-loop, mapping from the initial state distribution to a full sequence of actions, and operates as follows. Every demonstration is represented by the initial frame, i.e. the camera observations at the beginning of the demo, and a sequence of 3-D waypoints for the gripper motion. These waypoints, projected onto the image, identify important visual "keypoints" for that demo. To execute the policy starting from an initial observation of the scene, we identify the best-matched demo based on the quality of keypoint correspondences, backproject those keypoints to compute desired 3-D gripper waypoints, and warp the demonstrated trajectory based on those waypoints. We walk through these steps in detail below.

**Demonstration Summaries: Image, Waypoints, and Keypoints**. Our policy is non-parametric, and relies on accessing demonstrations at test time. For convenience, we first preprocess all demonstrations into concise summaries in preparation for execution. This is a "training time" operation, it needs to be done once for each demonstration, and once summarized, the original demonstration can be discarded. For each demonstrated trajectory $\tau_i \in \mathcal{D}$, we summarize the key information for Tether in a tuple $\kappa_i = (o, W, K, \mathbf{a})$, where $o$ is the initial image, i.e. two-view camera observations at the beginning of the trajectory. $W$ is a "waypoint" sequence $[w_1, w_2, ..., w_T]$ of task-critical 3-D gripper locations. In practice, we simply use the gripper locations from frames where the gripper open/close position toggles, following the convention for selecting critical frames from prior work (Johns, 2021; Mandlekar et al., 2023; Vecerik et al., 2024), and which is applicable to diverse manipulation tasks; in Appendix A.1, we also show that Tether works well with alternative waypoint extraction mechanisms. Next, $\mathbf{a} = [a_1, ..., a_M]$ is the full sequence of robot actions during the trajectory, i.e. 6-DOF gripper

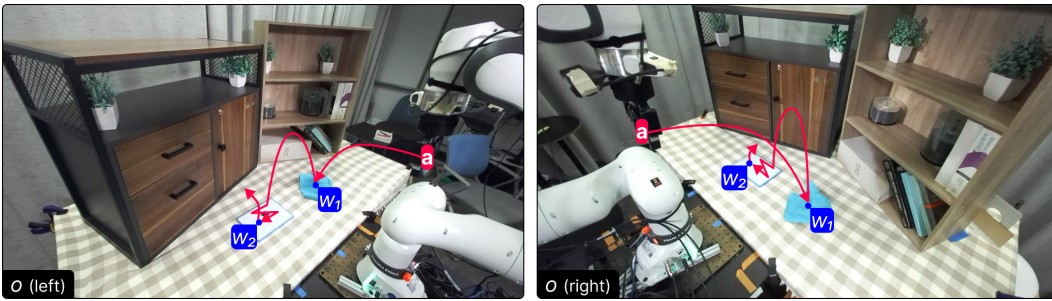

Figure 1: **Demonstration Summaries.** Tether summarizes demonstrations into the initial frame, action sequence (red), waypoints (blue), and keypoints.

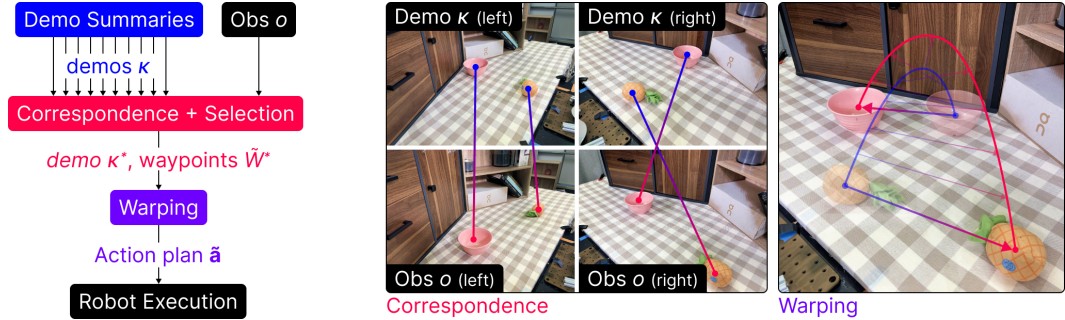

Figure 2: **Policy Inference**. During inference, Tether (left) computes correspondences (middle) and produces a warped trajectory action plan (right).

positions and gripper state. Finally, for each of the $T$ waypoint locations, we project them onto $o$ to identify visually important keypoints $K = [k_1, k_2, ..., k_T]$. We depict this process in Figure 1.

This remaining subsection outlines executing our policy for an observation $o$, visualized in Figure 2.

**Correspondence Matching and Source Demo Selection.** To execute a Tether policy, we start by matching the current camera observations $o$ to the demos in $\mathcal{D}$ to find a nearest-neighbor demo. In particular, for each demo summary $\kappa_i$, we search the current images $o$ for correspondences for all keypoint locations $K_i$ in the demo images $o_i$. We do this separately for the left-image and right-image to find a set of corresponding pixels in each image $\tilde{K}_i = [\tilde{K}_{i,left}, \tilde{K}_{i,right}]$. To find these 2D correspondences, we use a state-of-the-art model (Zhang et al., 2024) built on DINOv2 (Oquab et al., 2023) and Stable Diffusion (Rombach et al., 2022) features in our implementation.

We then backproject these correspondences $\tilde{K}_i$ using calibrated camera extrinsics to obtain a sequence of target 3-D waypoints $\tilde{W}_i$. If the backprojections fail to intersect, then the match is deemed to have failed, i.e. demo $i$ is an *infeasible* match for the current observation $o$. For the feasible matches, we rank the demos in order of dissimilarity to $o$ by computing the Euclidean distance between the original and target waypoints, i.e. $score_i(o) = \|W_i - \tilde{W}_i(o)\|_2$. The closest demo is selected as the source demo $\kappa^*$, with its original gripper waypoints $W^*$, original robot action sequence $\mathbf{a}^*$, and translated target waypoints for the current scene $\tilde{W}^*(o)$.

**Warping the Source Demo Trajectory.** The "target waypoints" $\tilde{W}^*(o)$ above provide a scaffold for how to warp the robot trajectory for the current scene. However, we still need to fill in the fine-grained robot actions in between by warping the intermediate segments between waypoints.

Consider the segment $[w_t, w_{t+1}]$ between two consecutive waypoints from the selected source demo $\kappa^*$. The target waypoints are $[\tilde{w}_t, \tilde{w}_{t+1}]$. Denote the action sequence segment for this waypoint as $\mathbf{a}_t^*$. We first compute the 3-D displacements of the two waypoints that mark the beginning and end of the segment, $d_t = \tilde{w}_t - w_t$ and $d_{t+1} = \tilde{w}_{t+1} - w_{t+1}$. We now perform linear interpolation between those displacements and add the resulting displacements to the original action sequence $\mathbf{a}_t^*$ to get the transformed action plan segment $\tilde{\mathbf{a}}_t$. Concatenating these segments produces the full action plan $\tilde{\mathbf{a}} = [\tilde{\mathbf{a}}_1, \tilde{\mathbf{a}}_2, ...]$ to be executed in the new scene.

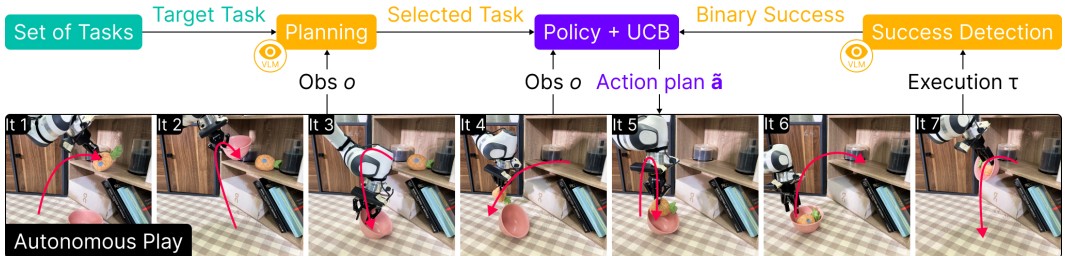

Figure 3: **Autonomous Functional Play.** Our iterative procedure runs Tether for multiple tasks and uses VLMs for plan generation and success detection.

To prioritize preserving spatial relationships, we perform this linear interpolation in space rather than time. For waypoints $w_t, w_{t+1}$, we define a local 1-D coordinate frame mapping $w_t$ to 0 and $w_{t+1}$ to 1. Then, the interpolation coefficient $\alpha$ for each $a \in$ the source action segment $\mathbf{a}_t^*$ is simply its coordinate in this frame. Geometrically, this can be thought of as projecting the gripper position $a$ onto the line spanning $w_t$ and $w_{t+1}$, then computing the projection's relative distance to $w_t$ and $w_{t+1}$. Then, the corresponding displacement that $a$ must undergo when warped into the new scene is: $d_a = (1 - \alpha)d_t + \alpha d_{t+1}$. The new action is thus $a + d_a$.

While simple, we show in Section 4.2 that our correspondence-driven trajectory warping performs remarkably well with as few as 10 demos, across challenging manipulation tasks—including those with out-of-distribution objects, millimeter-level precision, and complex contacts. Additional details and pseudocode are in Appendix.

## 3.2 Autonomous Functional Play with Vision-Language Models

Now, we address the scalability issue of human data collection by using our Tether policy to set autonomous play in motion. This expands the data distribution starting from a few demos per task, so that the expanded, more diverse dataset can power larger and more flexible neural policy architectures.

To maximize the autonomy of our data generation, we apply our policy towards continuous functional play and design a set of tasks that facilitate natural resets and randomization: the end state of each task is a valid start state for another task (e.g., "place pineapple on table" leads into "place pineapple on shelf" or "place pineapple in bowl"), even in the event of failures. This approximately indefinitely composable structure is an extension of forward-backward tasks in prior reset-free learning (Eysenbach et al., 2017; Sharma et al., 2021; Mirchandani et al., 2024), and promotes a set of reachable states that is effectively "closed" under the task distribution, while allowing previous tasks (and potential mistakes) to naturally randomize both relevant and background object locations and states for each task. This formulation is further illustrated by experiments in Section 4.3.

With these tasks, we run an iterative procedure where each step applies the policy to complete one task from our set. At a high level, each iteration proceeds as follows: we first query a VLM with an image of the scene and ask for an appropriate task to attempt. Then, we run the corresponding Tether policy, record its execution, and evaluate it with another VLM query. These steps are visualized in Figure 3 and described below, along with pseudocode in Appendix.

**Task Selection And Planning.** At each iteration, we select tasks based on both which demos we would like to add to our collection, and which tasks are actually executable. To prioritize demos to add, we maintain a running count of the number of successes for each task, and weight rare tasks higher. In particular, we sample the target task from a softmax over the negated success counts.

However, this rare target task might not be instantly executable. For instance, to attempt to "move object from shelf to table," the object of interest must have been placed on the shelf in the first place. If that object could be in many other locations in the scene, it might only rarely reach the shelf in the course of undirected play. To overcome this, we query a VLM to provide a task plan, i.e. a sequence of executable tasks that culminates in the target task, of which we attempt the first task within the current iteration, similar to receding horizon control. Our prompt and examples are in Appendix.

**Success Evaluation.** Having attempted a task, we determine the success of the resulting trajectory via a VLM query. We follow the recommended practice (Team, 2025), providing images of the

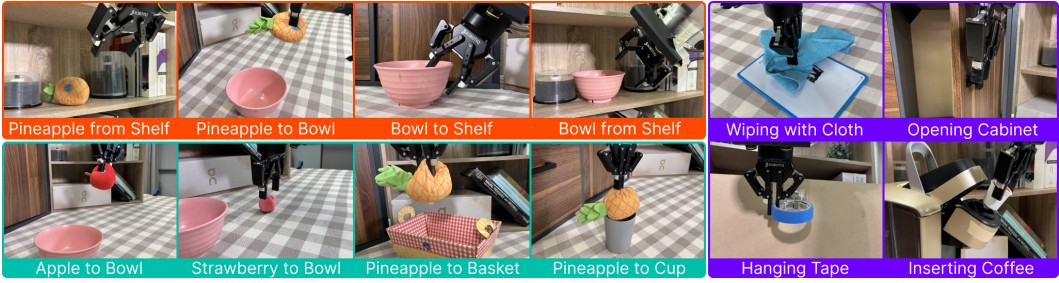

Figure 4: **Evaluation Tasks.** Our tasks involving moving fruits and containers with in-distribution (orange) and out-of-distribution (green) objects, as well as challenging manipulation skills (purple).

pre- and post-execution scene state from left, right, and wrist cameras. While prior work (Duan et al., 2024a) found that naively querying off-the-shelf VLMs results in bias toward false positives, we have observed near-perfect performance in our settings, possibly owing to the multi-camera full-observability setup and the use of a VLM trained specifically for embodied reasoning tasks (Gemini Robotics-ER 1.5 (Team et al., 2025)).

**Improving For and Through Play.** There are a few additional considerations when deploying Tether. First, play benefits from injecting stochasticity to generate exploratory data to search for improved policies. This also expands the set of potential actions, decreasing the chance of being trapped in states where all actions fail to induce environment transitions. Thus, rather than providing our policy with all demos, we first sub-select $k$ demos and run our policy to warp the closest one amongst them.

While we can select these $k$ demos randomly, there is a second consideration: human demos could be of varying quality, in which case, play using various source demos could help identify *consistently better demos* to warp from, avoiding e.g. demos that involve non-robust fingertip grasps. Thus, we select the top $k$ demos by formulating a multi-arm bandit problem: arms are demos, and the reward for picking a demo is the binary success of the executed trajectory warped from that demo. For this problem, we use upper confidence bounds (Garivier & Moulines, 2011), which balances exploring relatively less-tested source demonstrations with exploiting high-success ones.

## 4 EXPERIMENTS

We conduct experiments evaluating our policy design and autonomous play procedure. Specifically, we study (1) the robustness of our Tether policy to large, potentially out-of-distribution variations in the initial state configuration (i.e., new poses or new object instances), and (2) the effectiveness of autonomous play in generating a stream of data for downstream closed-loop policy learning.

### 4.1 EXPERIMENTAL SETUP

We run all experiments on the 7 DOF Franka Emika Panda arm running at 15 Hz and record two RGB views from calibrated ZED cameras. Across all experiments, we provide 10 demonstrations for each task. We run our semantic correspondence on 1 A6000 GPU. In autonomous play, we use Gemini Robotics-ER 1.5 for task selection and success evaluation. Additional details are in Appendix.

**Baselines And Ablations.** We compare our Tether policy with recent imitation learning methods. These baselines are representative of the state-of-the-art across various levels of data-efficiency. First, $\pi_0$ (Black et al., 2024) (with FAST (Pertsch et al., 2025)) is an open-source VLA that can operate zero-shot or finetuned with additional demos. Second, Keypoint Action Tokens (KAT) (Di Palo & Johns, 2024) queries a LLM for in-context action sequence generation, given a few demos (10 in the original work). Third, Diffusion Policy (DP) (Chi et al., 2023) is a general imitation learning algorithm typically trained with a few tens up to a few hundreds of demos, depending on task complexity. Here, we evaluate $\pi_0$ zero-shot and finetuned with 10 demos, and provide DP and KAT with 10 demos as well, same as our approach. Additionally, we ablate the number of demos given to our method, evaluating with 1, 5, and 10.

**Tasks.** We visualize our 12 tasks in Figure 4. First, we have 4 tasks that involve moving fruits and containers on a table and shelf. We instantiate these tasks with a soft pineapple toy and a curved rigid

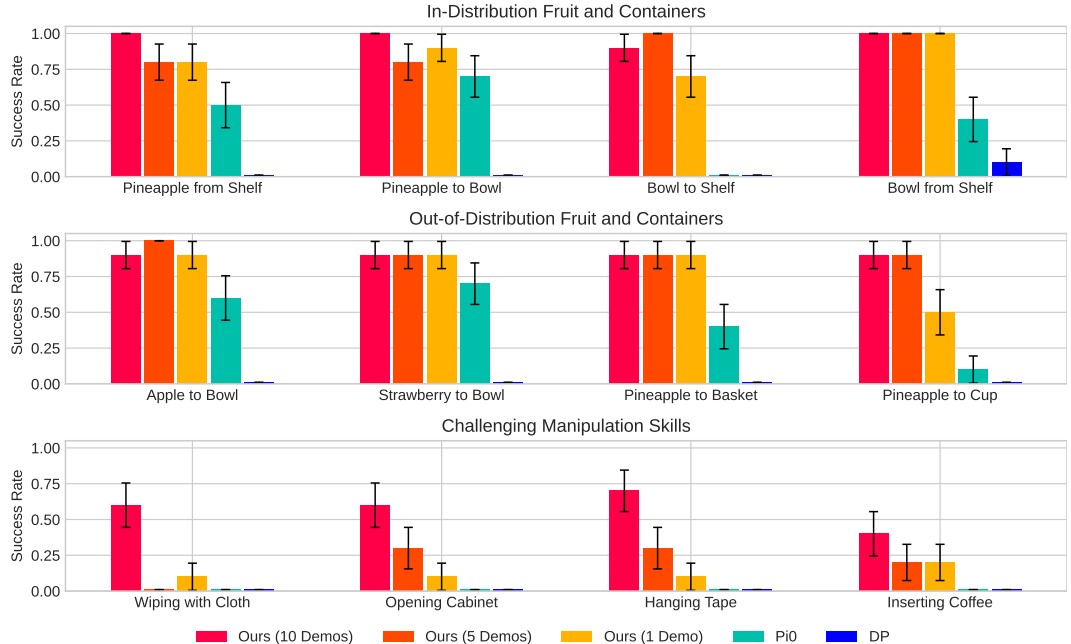

Figure 5: **Main Policy Comparison.** We compare the Tether policy with baselines across 12 tasks.

bowl. The main challenges are the bowl, which requires careful orientation of the gripper to prevent slipping, and the shelf, which requires a horizontal orientation approach to avoid collision.

We start with in-distribution objects (the same pineapple and bowl from demos) to specifically evaluate spatial generalization, since our few demos do not fully cover the entire distribution of object positions. Afterwards, we test with out-of-distribution objects to also assess semantic generalization: focusing on the "Pineapple to Bowl" task, we change the pineapple to an apple (color change) or strawberry (size change) and the bowl to a basket (appearance change) or cup (geometry change).

Next, we probe the limits of our policy design in 4 challenging tasks involving deformation, sustained contacts, articulation, and precision. First, the robot picks up a soft cloth and maintains consistent contact to wipe marks off a whiteboard. Second, it grasps a cabinet doorknob 0.5 centimeters thick (1/4 of gripper width) and applies stable contacts to fully open the tight hinge. Third, it places a roll of tape on a small silver hook, 3 centimeters deep and visible in only a few pixels. Fourth, it inserts a K-cup pod into a coffee machine, which demands precision with an error margin of 8 millimeters.

We report success rates over 10 trials. For each trial, we randomize object positions and use the same two camera views of the entire scene (in Figure 1). This naturally includes irrelevant locations in the majority of the image, requiring methods to focus on relevant regions of the scene. The "Inserting Coffee" task is the sole exception: since the 8mm margin of error projects onto the camera views as 2 to 3 pixels, we move the cameras closer to zoom in on the coffee pod and machine compartment.

## 4.2 ROBUST IMITATION

**Tether policy outperforms alternative imitation learning approaches.** In Figure 5, we find that given few demos, our policy surpasses baselines across all tasks. Diffusion Policy, being an end-to-end model trained from scratch without built-in priors, fails to generalize from just 10 demos. Meanwhile, zero-shot $\pi_0$ performs well on standard tabletop pick-and-place, likely benefiting from its pretraining on datasets containing similar behaviors, but fails with more complex tasks due to (1) incomplete command understanding (e.g., grasping the cloth but failing to locate the whiteboard) and (2) imprecise manipulation (e.g., approaching the coffee pod but missing the grasp).

Next, two baselines failed to achieve any successes on our tasks. First, we observe that finetuned $\pi_0$ collapses when trained on only 10 demos, likely due to severe overfitting. When deployed, this model often does not move at all, or misses the object. Second, our few-shot learning baseline, KAT, was not well-suited for extracting task-relevant features from our cluttered scene, and even when given human-

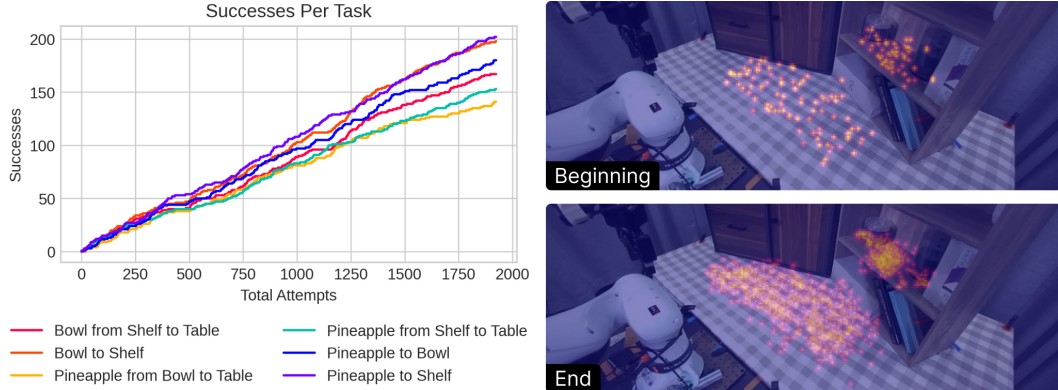

Figure 6: **Autonomous Play Statistics.** In around 26 hours of play, our method produces over 1000 cumulative trajectories across 6 tasks (left) and significantly expands data diversity, as shown by the heatmaps of object poses at the beginning and end of play (right).

annotated inputs, it struggled with in-context learning from the complex multi-dimensional patterns in our tasks, caused by orientation changes, non-linear velocities, and wide object distributions. Additional details and experimental verifications for both baselines are in Appendix.

**Tether policy excels at both spatial and semantic generalization.** First, our policy excels in tasks involving spatial robustness with in-distribution objects, including the bowl, which requires accurate orientation and position to avoid slipping. Second, our policy performs well even with out-of-distribution objects, attesting to the strong generalization inherited from visual correspondence: without the demo object being present at test-time, correspondence finds the most semantically similar object and pinpoints the relative region of the keypoint (e.g., center of fruit or rim of container). This capability is significant especially for grasping the strawberry, which is vastly different in appearance and 1/4th the size of the demonstrated pineapple, and cup, which has 1/2 the diameter of the bowl and just big enough to fit the pineapple, thus requiring precision along with generalization.

**Tether policy succeeds even on challenging manipulation tasks.** Our policy is effective at more difficult manipulation tasks despite their challenges with deformation, sustained contacts, articulation, and precision. Successes with small features of the scene like the cabinet knob, hook, and coffee machine compartment demonstrate the high accuracy of our semantic correspondences. Most notably, our method achieves non-trivial success for the coffee insertion task without using the wrist camera; this task requires significant accuracy during grasping, since the deformable cylindrical pod crumbles if held just 1 centimeter off, and insertion, since the difference in pod and compartment diameter is a mere 8 millimeters. Additionally, fully opening the cabinet and wiping marks off the whiteboard show that trajectory warping can maintain consistent contacts with the scene, even without closed-loop adjustments. Lastly, accurate grasps of the soft cloth demonstrate our method's flexibility with deformable objects, in contrast with prior work (Wen et al., 2022; Mandlekar et al., 2023; Zhu et al., 2024) that rely on rigid objects for pose estimation.

Finally, while our Tether policy exhibits strong robustness to spatial and semantic variations in the initial state, even for challenging tasks, its open-loop nature remains a bottleneck, limiting reactivity and recoveries during execution. Thus, it serves not as an independent policy solution, but as a bootstrap for generating experience and training more expressive policies, as we investigate below.

### 4.3 AUTONOMOUS PLAY

Given our policy's strong performance, we now run a subset of evaluation tasks for autonomous play. Specifically, we use the demos from Section 4.2 (10 per task) for moving the pineapple between table, shelf, and bowl, as well as moving the bowl between table and shelf. As described in Section 3.2, this task set is chosen so that there is a high certainty of at least one viable subsequent task that can be executed after previous tasks, even in their most common failure cases (e.g. object drops to table).

**Tether runs across multiple hours without resets.** We run autonomous functional play for 4 sessions totaling around 26 hours. Success statistics for each task are shown in Figure 6 (left). Our policy generates 1085 successes from 1946 attempts across our 6 tasks, averaging 1 success every 86

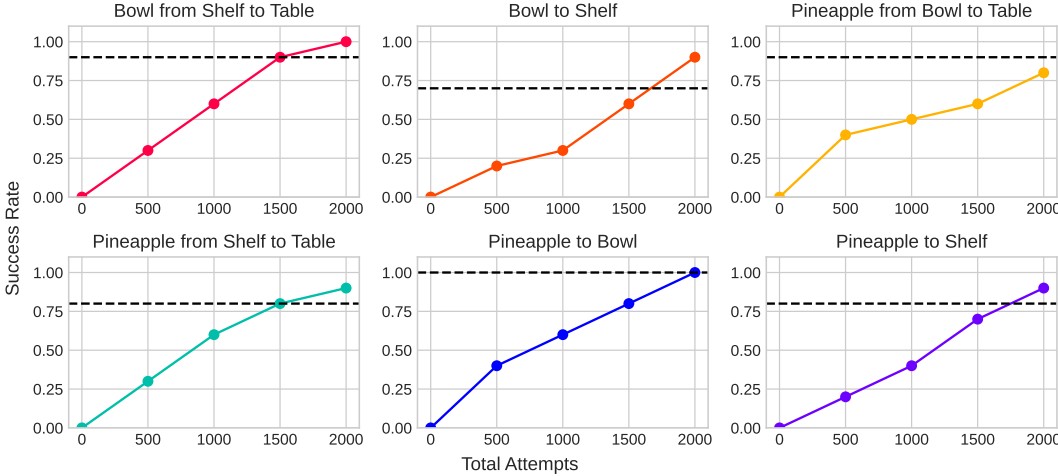

Figure 7: **Downstream Policy Learning Results.** The stream of data generated by autonomous play consistently improves policy performance over time. Policies trained on human-collected demos, equal in number to the final set of successful Tether-generated trajectories, are shown in black.

seconds and 1 attempt every 48 seconds, with a cumulative success rate of 55.8%. We intervene a total of 5 times, amounting to 0.26% of attempts and an average of once every 5.2 hours. Timelapse and intervention details are in Appendix.

We observe that these success rates are lower than those in Section 4.2 due to the uncontrolled nature of play. For instance, the bowl is sometimes tilted on its side due to imprecise placements or accidental pushes; these make grasps significantly harder, though our policy is still able to recover and fix the bowl. On rare occasions, play enters a failure mode where the bowl is flipped completely upside-down after dropping from the shelf; this state is generally irrecoverable with only one arm and causes the majority of our 5 interventions. However, on two separate instances, the robot accidentally recovers by squeezing the bowl against the shelf and forcing it back upright. While such recoveries are not intended and occur purely by luck, they highlight the interesting nature of play: that at scale, coincidences may result in unexpected novel behaviors.

**Tether produces reliable task plans and success evaluations.** We evaluate our VLM-based planning and evaluation by annotating all 1946 attempts: we deem a task plan as correct if it can be feasibly executed given the current scene image, and an execution as successful if it achieves the task's end state without disturbing task-irrelevant objects. Compared against the VLM responses, we achieve 95.2% accuracy for task planning and 98.4% precision (at 89.6% recall) for success evaluation; note that we prioritize precision as it's most important to minimize false positives that pollute the success data. These results validate the reliability of our VLM-based components for play.

**Tether produces diverse trajectories.** In Figure 6 (right), we visualize the diversity of keypoints from demonstrations and successful play trajectories. We see that while our demos sparsely cover the table and shelf, using them to seed play allows us to not only interpolate between the demos but also expand on the edges of the distribution, such as the area around the cabinet.

**Tether produces a stream of data that trains effective policies.** Next, we train parametric policies on the generated data. While there are numerous algorithms for learning from suboptimal data, we adopt filtered behavioral cloning as a straightforward and effective approach. Integrating alternative methods that make fuller use of suboptimal trajectories remains a key direction for future work.

After every 500 play attempts, we train Diffusion Policies on the cumulative successful trajectories for each task. We track their performance in Figure 7. Across all tasks, these policies progressively improve over time, with most eventually reaching near-perfect success rates. Thus, the data generated by Tether is consistently high-quality and effective for downstream policy learning. Diving deeper into the nature of this improvement, we see that as our ever-expanding datasets scale with more play, they increase in diversity and naturally build a stronger coverage of the object pose distribution. Thus, policies trained on this data improve primarily on their spatial robustness to different object positions: earlier policies succeed only when the object is in some specific locations, whereas our final policies are generally effective for any random object placement. Note that this robustness also extends to

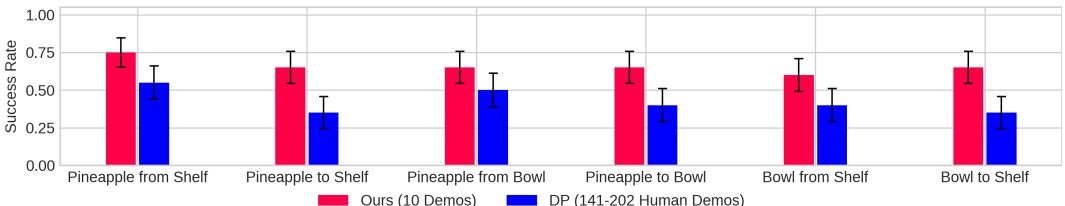

Figure 8: **Comparison on Play Distribution.** We compare Diffusion Policies trained on human-collected demos with Tether policy on environment states encountered during autonomous play.

distractor objects that were involved in other tasks during play: for instance, policies that interact only with the bowl perform well irrespective of the pineapple position, and vice versa.

Additionally, we compare against policies trained on human-collected datasets, using the same number of human demos as the final set of successful trajectories produced by Tether (between 141 and 202, depending on task). As seen in Figure 7, our policies generally achieve success rates similar to their human-data counterparts, thus confirming that Tether indeed produces expert-level trajectories competitive with those of experienced teleoperators, while requiring minimal human effort. On average across the 6 tasks, our policies even achieve slightly higher success rates, which we hypothesize arises from two factors. First, the environment is naturally randomized when play is performed at scale, as opposed to potentially biased randomization during manual resets. Second, Tether trajectories are produced by warping from a small demo set, and thus its motions occupy a relatively narrow (yet effective) mode of the expert distribution, whereas the hundreds of human-collected motions are relatively more multimodal and thus more difficult for the policy to capture.

**Tether policy is essential for robust autonomous play.** Finally, we investigate whether the Tether policy is only valuable when the number of source demos is very small, and if standard imitation learning approaches are more effective if we have more demos. To answer this, we simulate the result of plugging in the above human-data Diffusion Policies into play, replacing the Tether policy: for each of the 6 tasks, using 20 randomly sampled play-induced initial states from our multi-hour play experiment, we compare Tether policy (with 10 demos) and Diffusion Policy (with 141 to 202 demos, depending on task). As seen in Figure 8, Tether policy consistently achieves higher success rates, whereas Diffusion Policies fail to generalize to the broader play state distribution (e.g. tilted bowl or entangled objects). Thus, our policy design is key to sustaining efficient and robust autonomous play, minimizing human intervention and maximizing throughput.

## 5 CONCLUSION

We have presented Tether, a system for autonomous functional play with structured, task-directed interactions. We introduce a novel policy design with keypoint correspondence and trajectory warping that exhibits impressive spatial and semantic robustness, and we deploy it within a VLM-guided multi-task play procedure that produces over 1000 successful trajectories in 26 hours with minimal human intervention. This generated data, funneled downstream for filtered imitation learning, consistently improves the performance of neural policies, which ultimately reach near-perfect success rates. We believe that Tether demonstrates the potential for an alternative path in robot learning: one driven by scalable methods that perform and learn from autonomous interaction and experience.

**Limitations.** Our policy's strength in exploiting semantic correspondences is also a weakness: recent advances in computer vision enable robust feature tracking despite textureless surfaces, deformable regions, lighting variations, and partial occlusions, yet the abstraction of an image into keypoints inherently makes the policy more susceptible to occlusions. In addition, our policy is deliberately designed for the low-data regime, and thus has built-in structural generalization and assumptions that are not suitable for larger datasets; in particular, open-loop execution limits its applicability to dynamic tasks that require reactivity, and trajectory warping struggles with complex motions that cannot be transformed from source demos. A promising future direction could leverage the Tether policy as a strong prior, while remaining flexible to improving via imitation or reinforcement learning as more data becomes available through play, thus enabling more effective self-improvement.

ACKNOWLEDGMENTS

We are grateful to Jie Wang, Edward Hu, Junyao Shi, and Pieter Abbeel for their helpful feedback and discussions during the project. William Liang is supported by the NSF GRFP. This project was funded by NSF SLES 2331783, NSF CAREER 2239301, ONR N00014-22-1-2677, and DARPA TIAMAT HR00112490421.

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

# A APPENDIX

## A.1 TRAJECTORY WARPING

**Pseudocode.** In Algorithm 1, we describe our policy method in detail.

**Alternative Waypoint Extraction Mechanisms.** In our main results, we perform waypoint extraction by identifying frames where the gripper open/close position toggles. However, Tether is agnostic to the specific choice of waypoint extraction method, which can be readily exchanged for alternatives.

For illustration, we propose the example task of "Pouring from Cup to Bowl," which is difficult because (1) it requires a waypoint at the pouring location, which cannot be extracted via gripper change as the gripper is holding the cup, and (2) the keypoint used for corresponding the bowl does not lie at the gripper position, which is hovering above the bowl. In this case, we instead query a VLM to select keyframes from the demo video and point to the 2D pixel for the waypoint; then, we triangulate the pixels from both views to compute the waypoint. Finally, we continue the original algorithm to run correspondence and warp the trajectory. We empirically verify this alternative mechanism in a real world setup, where our method successfully identifies waypoints for the cup and bowl and achieves 90% success.

**Correspondence Filtering.** As explained in Section 3.1, our primary method of filtering out invalid demo correspondences is triangulation: if the backprojected correspondences do not intersect within a threshold of 10 centimeters, then we deem the demo as an *infeasible* match for the observation $o$.

However, we find in practice that incorrect correspondences may have a valid triangulation purely by coincidence, thereby passing the feasibility check. In the worst case, this can cause dangerous collisions. As our robot setup lacks built-in collision avoidance, we opt to build a simple custom safety layer via the fast multi-view correspondence algorithm MAST3R (Leroy et al., 2024). For both the demo and observation $o$, we match each keypoint from its view to the same scene in the other camera view (left to right camera, and vice versa). We then compute the distance of the 3-D waypoint to the ray projected by this match, and if the distance differs between the demo and observation by over 10 centimeters, we deem this correspondence as infeasible as well.

**Post-Warp Speed Adjustment.** Another key consideration in implementing trajectory warping is the difference in relative waypoint positions $w_t, w_{t+1}$ between the demo and warped trajectory: if there is a large position difference (e.g. objects closer in demo but farther apart in observation) and we keep the number of intermediate timesteps constant, then the resulting warped trajectory has a large velocity difference as well, which can be dangerous.

Thus, after the warping process described in Section 3.1, we recompute intermediate timesteps such that velocity remains the same between demo and warped trajectories. Specifically, we scale the number of intermediate keypoints $T_{\text{segment}}$ by the arc length $\tilde{L}$ of the new action plan segment $\tilde{\mathbf{a}}_t$ relative to the arc length $L$ of the original segment $\mathbf{a}_t$, $\tilde{T}_{\text{segment}} = (\tilde{L}/L)T_{\text{segment}}$. Then, we compute new intermediate actions following the relative distribution of the original intermediate actions: for original intermediate action $a_t$, we map timestep $t/T_{\text{segment}}$ to $s/L$ where $s$ is the arc length from $w_t$ to $a_t$. Then, we interpolate along this function to map new timesteps $t/\tilde{T}_{\text{segment}}$ to $\tilde{s}/\tilde{L}$, and compute the resulting intermediate action as the position with $\tilde{s}$ arc length to $\tilde{w}_t$.

## A.2 AUTONOMOUS PLAY

**Pseudocode.** In Algorithm 2, we describe our play procedure in detail.

**Multi-Task Design.** Our 6 tasks for autonomous play, as plotted in Figure 6, are as follows:

1. Place pineapple from table to shelf.
2. Place pineapple from shelf to table.
3. Place pineapple from table to bowl.
4. Place pineapple from bowl to table.
5. Place bowl from table to shelf.
6. Place bowl from shelf to table.

---

**Algorithm 1** Trajectory Warping with Keypoint Correspondences

---

1: **Require**: Observation $o$, Demos $D$
2: **for** $\kappa_i = (o_i, W_i, K_i, \mathbf{a}_i) \in D$ **do**
3: $\quad$ // Find keypoint correspondences between demo and observation
4: $\quad$ $\tilde{K}_i \leftarrow \text{findCorrespondences}(K_i, o_i, o)$
5: $\quad$ // Backproject correspondences to compute waypoints
6: $\quad$ $\tilde{W}_i \leftarrow \text{backprojectStereo}(\tilde{K}_{i,left}, \tilde{K}_{i,right})$
7: **end for**
8: // Select the closest-matched demo
9: $i^* \leftarrow \arg\min_i score_i(o)$
10: $W^*, \tilde{W}^*, \mathbf{a}^* \leftarrow W_{i^*}, \tilde{W}_{i^*}, \mathbf{a}_{i^*}$
11: // Warp each action segment
12: **for** $t = 1, \ldots, T-1$ **do**
13: $\quad$ // Compute waypoint displacements at start and end of segment
14: $\quad$ $d_t \leftarrow \tilde{w}_t - w_t$
15: $\quad$ $d_{t+1} \leftarrow \tilde{w}_{t+1} - w_{t+1}$
16: $\quad$ **for** $a \in \mathbf{a}_t^*$ **do**
17: $\quad\quad$ // Compute linear interpolation in space
18: $\quad\quad$ $v \leftarrow w_{t+1}^* - w_t^*$
19: $\quad\quad$ $\alpha \leftarrow (a - w_t^*)^\top v / \|v\|^2$
20: $\quad\quad$ $d_a \leftarrow (1 - \alpha)d_t + \alpha d_{t+1}$
21: $\quad\quad$ $\tilde{a} \leftarrow a + d_a$
22: $\quad$ **end for**
23: $\quad$ $\tilde{\mathbf{a}}_t \leftarrow [\tilde{a} \mid a \in \mathbf{a}_t^*]$
24: $\quad$ // Compute post-warp speed adjustment
25: $\quad$ $\tilde{\mathbf{a}}_t \leftarrow \text{adjustSpeed}(\tilde{\mathbf{a}}_t)$
26: **end for**
27: **Output**: Warped action plan $\tilde{\mathbf{a}} = [\tilde{\mathbf{a}}_1, \tilde{\mathbf{a}}_2, \ldots]$

---

**Algorithm 2** Autonomous Functional Play with Vision-Language Models

---

1: **Require**: Tether Policy $\pi$, Task Library $T$, Demos $\{\mathcal{D}_t\}_{t \in T}$, Success Detector $\mathcal{S}$, Task Planner $\text{VLM}_{planner}$, Success Evaluator $\text{VLM}_{evaluator}$
2: **Hyperparameters**: Demo subsample size $k$
3: // Initialize generated trajectory set $\mathcal{G}_t$ for each task $t$
4: $\{\mathcal{G}_t \leftarrow \emptyset\}_{t \in T}$
5: **while** TRUE **do**
6: $\quad$ Receive observation $o$
7: $\quad$ // Select a task $t_{targ}$ to target
8: $\quad$ $t_{targ} \sim \text{Softmax}(-|\mathcal{G}_t|)$
9: $\quad$ // Generate a plan for $t_{targ}$ and attempt first task $t$
10: $\quad$ $\{t, \ldots\} = \text{VLM}_{planner}(o, t_{targ})$
11: $\quad$ // Select $k$ demos for $t$ using UCB
12: $\quad$ $\mathcal{D} \leftarrow \text{TopK}_k (\text{UCB}(\kappa_i) \mid \kappa_i \in \mathcal{D}_t)$
13: $\quad$ // Run Tether to select demo $\kappa^*$ and produce rollout $\tau$
14: $\quad$ $\tau, \kappa^* = \pi(\mathcal{D}, o)$
15: $\quad$ // Check success and update UCB
16: $\quad$ **if** $\mathcal{S}(\tau, \kappa^*, \text{VLM}_{evaluator})$ **then**
17: $\quad\quad$ // If successful, add $\tau$ to generation set $\mathcal{G}_t$
18: $\quad\quad$ $\mathcal{G}_t \leftarrow \mathcal{G}_t \cup \{\tau\}$
19: $\quad\quad$ $r \leftarrow 1$
20: $\quad$ **else**
21: $\quad\quad$ $r \leftarrow 0$
22: $\quad$ **end if**
23: $\quad$ $\text{UCB}(\kappa^*) \leftarrow \text{UpdateUCB}(\text{UCB}(\kappa^*), r)$
24: **end while**
25: **Output**: Generated trajectories $\{\mathcal{G}_t\}_{t \in T}$

---

This set of tasks possesses the approximately indefinitely composable structure as described in Section 3.2: regardless of execution success or failure, the pineapple will be at the table, shelf, or bowl, and the bowl will be at the table or shelf. Then, for any of these combinations as a start state, there is at least one task from the list above that is valid.

**VLM Queries.** Below, we present the VLM prompts and example responses from our play procedure.

```
Analyze this robot environment, captured from left and right camera views. Based on this scene
    , please create an action plan that ends with the task: <task>. If the task is already
    achieved in the scene, you should disrupt the scene so the task is not achieved anymore,
    and then plan to achieve the task.

The following atomic actions are available:
<actions>

Please provide your final answer in the following format:
```json
{
    "description": "<description of the scene>",
    "actions": [
        {
            "action": "<action to execute>",
            "action_letter": "<list letter>",
            "reasoning": "<reasoning for action>"
        },
        ...
    ]
}
```
```

Listing 1: Task Plan Generation.

```
```json
{
    "description": "The scene shows a robot arm positioned near a table covered with a
        checkered tablecloth. On the table, there is a pink bowl containing a toy pineapple.
        A wooden shelf unit and a cabinet are visible in the background.",
    "actions": [
        {
            "action": "pick up pineapple from bowl on table and place on table",
            "action_letter": "d",
            "reasoning": "The goal is to pick up the pineapple from the table and place it in
                the bowl. The current scene shows the pineapple already inside the bowl. To
                make the task achievable, we must first disrupt the current state by removing
                 the pineapple from the bowl."
        },
        {
            "action": "pick up pineapple from table and place in bowl on table",
            "action_letter": "c",
            "reasoning": "After removing the pineapple from the bowl and placing it on the
                table, we execute the final task of picking up the pineapple from the table
                and placing it back into the bowl."
        }
    ]
}
```
```

Listing 2: Task Plan Generation Example Response.

```
For this task, you will see a robot trying to perform the task of <task>. You may see multiple
     camera views of the same scene. Some cameras are static and are mounted outside of the
    scene and some cameras are mounted on the robot arms and thus they are moving during the
    episode.
The first 3 images show multiple camera views from the start of the episode (some time ago).
    The last 3 images show multiple camera views from the current moment in the episode (as
    it is now).
Looking at these images and comparing the start of the episode with current state did the
    robot successfully perform the task "<task>"? Make sure that the task is completed
    without disrupting other objects in the scene.

Please think step by step and provide your reasoning. Then, provide your final answer in the
    following format:
```

```json
{
    "reasoning": <reasoning for evaluation>,
    "completed": <true or false>
}
```

Listing 3: Success Evaluation.

```json
{
    "reasoning": "Step-by-step analysis:\n1.  **Initial State Analysis (Images 1-3):** The
        pink bowl is located on the checkered surface (table area) in front of the wooden
        shelf unit. The middle shelf of the wooden unit is visible and does not contain the
        bowl.\n2.  **Current State Analysis (Images 4-6):** The pink bowl is no longer on the
         checkered surface. Instead, it is now placed on the middle shelf of the wooden unit.
         The robot arm's gripper is positioned near the bowl on the shelf, indicating it has
         either just placed it or is in the process of releasing it.\n3.  **Task Evaluation:**
         The task was to 'pick up bowl from table and place it on middle shelf'. The
        comparison between the initial and current images confirms that the bowl has been
        successfully moved from the table to the middle shelf. No other objects appear to
        have been disrupted during this process.",
    "completed": true
}
```

Listing 4: Success Evaluation Example Response.

**Correspondences for Success Evaluation** As explained in Section 3.2, we simply query Gemini Robotics-ER 1.5 to determine success. However, before the release of this model, we required an additional mechanism that reduces false positive rates by double-checking the VLM evaluation with a correspondence-based heuristic: from the final camera image $o^*_{final}$ of the source demo $\kappa^*$ that produced this trajectory, we look for correspondences of its final keypoint $k^*_{final,T}$ in the Tether execution's final image $o_{final}$. If one (or both) of the rays projected by the best-matched point from either camera view is far from the executed gripper position (beyond a preset threshold of 10 centimeters), we mark the trajectory a failure. Otherwise, if both the correspondence and VLM response are positive, then the trajectory is a success.

This correspondence-based verification is necessary to avoid false positive evaluations from our initial experiments using GPT-4.1, but is now largely redundant. Following the procedure outlined in Section 4.3, our post-hoc evaluation of Gemini Robotics-ER 1.5 without correspondence-based verification reaches 98.4% precision (at 89.6% recall), surpassing our previous results with GPT-4.1 and the correspondence-based verification, which reaches 97.7% precision (at 87.7% recall). Similarly, Gemini Robotics-ER 1.5 achieves 95.2% accuracy for task planning, surpassing our previous results with Gemini-2.5-Flash, which achieves 94.1%. Thus, while our play experiments were initially conducted with this mechanism and past VLM models, it is strictly more advantageous to simply adopt the most recent VLM advancements.

**Failure Modes.** While we only had 5 intervention-necessitating failures over 26 hours, the key failure mode of Tether is caused by the bowl flipping completely upside-down. It is nearly impossible to recover from this state with only a single arm, as un-flipping the bowl requires two separate points of contact. Besides such cases, other failure modes that we discovered and resolved in previous iterations of play experiments include crashing into the furniture, which we minimize via a simple out-of-bounds check, and dropping objects to the floor, which we avoid by placing barriers on the boundary of the robot's workspace.

## A.3 EXPERIMENTAL SETUP

**Diffusion Policy (DP) Setup.** For training and deploying all diffusion policies, we use the codebase provided by DROID (Khazatsky et al., 2024), following the original hyperparameters with a few modifications. Specifically, we set the batch size to 32, the shuffle buffer size to 1000, and training epochs to 600. For each task, we ablate the choice of camera views given to the policy: "Pineapple to Shelf" and "Pineapple from Shelf" use the left and right external cameras, while the rest use the wrist and right external cameras.

**Keypoint Action Tokens (KAT) Setup.** Keypoint Action Tokens (KAT) receives a series of teleoperated robot demonstrations, encodes the initial visual observation and actions into a pair of input and output tokens, and feeds the demo tokens and current visual observation to a LLM to generate a trajectory in the new scene. We find that KAT failed to achieve any successes on our tasks, despite ablating its three key components: keypoint tokens, action tokens, and the LLM used for queries. We review these attempted fixes below.

1. Keypoint tokens are extracted by extracting visual descriptors using DINO-ViT, extracting correspondences using Best-Buddy Nearest Neighbor matching between the descriptors of two initial observations in the input set of demonstrations, and then finding the corresponding location of each selected visual descriptor in the new scene. However, since our camera views capture the entire workspace, we find that these visual tokens often fall on parts of the scene that are constant across demos but are not relevant to the demonstrated task (e.g., cabinet in background); in contrast, locations that are crucial to track for the current task (e.g., pineapple) are not tracked by any of the descriptors. Thus, KAT may be susceptible to visual distractors. While the original paper experiments with distractors at test time, they are not present in the demos, and thus do not affect visual description extraction. To address this limitation, we manually select the relevant keypoints to track in an image. For example, for the task of picking up a pineapple and putting it into a bowl, we annotate keypoints on the location of the pineapple and bowl. However, we ultimately find that even with only the task-relevant keypoints marked, the output trajectory is not accurate.

2. In the original paper, KAT records actions at 4 Hz. We experimented with various frequencies between 3 Hz and 15 Hz, but find that none work well. Additionally, we observe that after the frequency increases beyond 10 Hz, the model frequently outputs the same trajectory, regardless of the keypoint locations for the input visual observation.

3. KAT originally uses GPT-4-Turbo to generate the new trajectory. We tested GPT-4-Turbo, but found that it often outputs invalid action sequences or inaccurate actions. We also tested with GPT-4.1, Gemini-2.5-Flash, and Gemini Robotics-ER 1.5 and find that while these models output validly formatted action sequences, the output action sequences are not accurate behaviors.

In summary, we believe that KAT's limitations are due to the failure of LLMs to handle multi-dimensional numerical patterns in our trajectories, caused by orientation changes, non-linear velocities, and wide object distributions. To verify this hypothesis, we run additional experiments with KAT on two tasks with simpler trajectory distributions: lifting the lid handle of a coffee machine, and opening a drawer. KAT achieves 70% and 60% (while Tether achieves 90% and 100%), which confirms that it indeed works well on more local object-centric interactions. This validates that while our few-shot imitation baseline succeeds on certain simpler trajectory distributions, Tether excels at handling more diverse, challenging trajectories.

**Finetuned $\pi_0$ Setup.** We finetune $\pi_0$ using the official codebase and checkpoints provided by Physical Intelligence. We finetune the $\pi_0$-FAST-DROID checkpoint for 1000 steps on the set of 10 human collected demonstrations we provide to Tether. During training and evaluation, we set the language prompt to the corresponding action description used in our VLM queries during play. We observe that training the model on a set of 10 demonstrations collapses the model, as it often does not move or misses the target object, while the model running zero-shot inference achieves nonzero success rates on the same tasks. However, to validate that this setup does indeed work for $\pi_0$ finetuning, we perform additional experiments finetuning $\pi_0$ on our Tether-generated data, with many more demos than the 10 human-provided ones for the original baseline. Details and results are in Section A.5 below.

## A.4 EXPERIMENTAL RESULTS

**Policy Comparison.** In Table 1, we report the exact figures from Figure 5 along with standard errors.

**Intermediate Tether Visualizations.** We visualize intermediate steps of our Tether policy from the evaluations in Section 4.2. Specifically, we include annotations for semantic correspondence, where the red dot indicates the correspondence match between source (demo) and target (scene), and the blue dot is the MAST3R match between viewpoints. Additionally, we include a few visualizations of

| Method | Pineapple from Shelf | Pineapple to Bowl | Bowl to Shelf | Bowl from Shelf |
|---|---|---|---|---|
| Tether (10 Demos) | $1.0 \pm 0.0$ | $1.0 \pm 0.0$ | $0.9 \pm 0.09$ | $1.0 \pm 0.0$ |
| Tether (5 Demos) | $0.8 \pm 0.13$ | $0.8 \pm 0.13$ | $1.0 \pm 0.0$ | $1.0 \pm 0.0$ |
| Tether (1 Demo) | $0.8 \pm 0.13$ | $0.9 \pm 0.09$ | $0.7 \pm 0.14$ | $1.0 \pm 0.0$ |
| $\pi_0$ (Black et al., 2024) | $0.5 \pm 0.16$ | $0.7 \pm 0.14$ | $0.0 \pm 0.0$ | $0.4 \pm 0.15$ |
| DP (Chi et al., 2023) | $0.0 \pm 0.0$ | $0.0 \pm 0.0$ | $0.0 \pm 0.0$ | $0.1 \pm 0.09$ |

| Method | Apple to Bowl | Strawberry to Bowl | Pineapple to Basket | Pineapple to Cup |
|---|---|---|---|---|
| Tether (10 Demos) | $0.9 \pm 0.09$ | $0.9 \pm 0.09$ | $0.9 \pm 0.09$ | $0.9 \pm 0.09$ |
| Tether (5 Demos) | $1.0 \pm 0.0$ | $0.9 \pm 0.09$ | $0.9 \pm 0.09$ | $0.9 \pm 0.09$ |
| Tether (1 Demo) | $0.9 \pm 0.09$ | $0.9 \pm 0.09$ | $0.9 \pm 0.09$ | $0.5 \pm 0.16$ |
| $\pi_0$ (Black et al., 2024) | $0.6 \pm 0.15$ | $0.7 \pm 0.14$ | $0.4 \pm 0.15$ | $0.1 \pm 0.09$ |
| DP (Chi et al., 2023) | $0.0 \pm 0.0$ | $0.0 \pm 0.0$ | $0.0 \pm 0.0$ | $0.0 \pm 0.0$ |

| Method | Wiping with Cloth | Opening Cabinet | Hanging Tape | Inserting Coffee |
|---|---|---|---|---|
| Tether (10 Demos) | $0.6 \pm 0.15$ | $0.6 \pm 0.15$ | $0.7 \pm 0.14$ | $0.4 \pm 0.15$ |
| Tether (5 Demos) | $0.0 \pm 0.0$ | $0.3 \pm 0.14$ | $0.3 \pm 0.14$ | $0.2 \pm 0.13$ |
| Tether (1 Demo) | $0.1 \pm 0.09$ | $0.1 \pm 0.09$ | $0.1 \pm 0.09$ | $0.2 \pm 0.13$ |
| $\pi_0$ (Black et al., 2024) | $0.0 \pm 0.0$ | $0.0 \pm 0.0$ | $0.0 \pm 0.0$ | $0.0 \pm 0.0$ |
| DP (Chi et al., 2023) | $0.0 \pm 0.0$ | $0.0 \pm 0.0$ | $0.0 \pm 0.0$ | $0.0 \pm 0.0$ |

Table 1: Success rates and standard error measured over 10 trials.

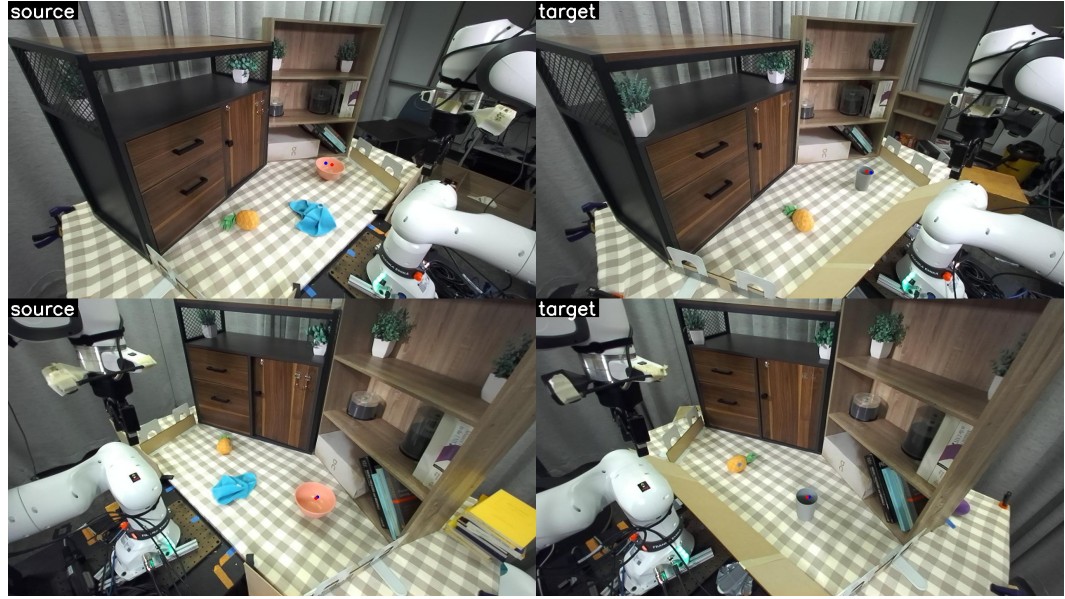

Figure 9: Computed correspondence for the Pineapple to Cup task.

our trajectory warping. Note that the demonstration depicted is the one selected by our method as the most similar to the scene.

In Figures 9 and 10, we visualize examples of successful semantic generalization: from the pineapple to strawberry and bowl to cup. These demonstrate the key ability of semantic correspondence to generalize beyond the demonstration, enabling interactions with out-of-distribution scenes.

In Figures 11 and 12, we visualize examples of successes in hanging tape and wiping the whiteboard with cloth. The former shows the accuracy of correspondence even with small objects like the silver hook, and the latter demonstrates correspondence with deformable, non-rigid objects like the cloth.

Finally, in Figures 13 and 14, we visualize examples of the full warped trajectory, with the source demo on the left, the warped result on the right, and annotated waypoints along the trajectory.

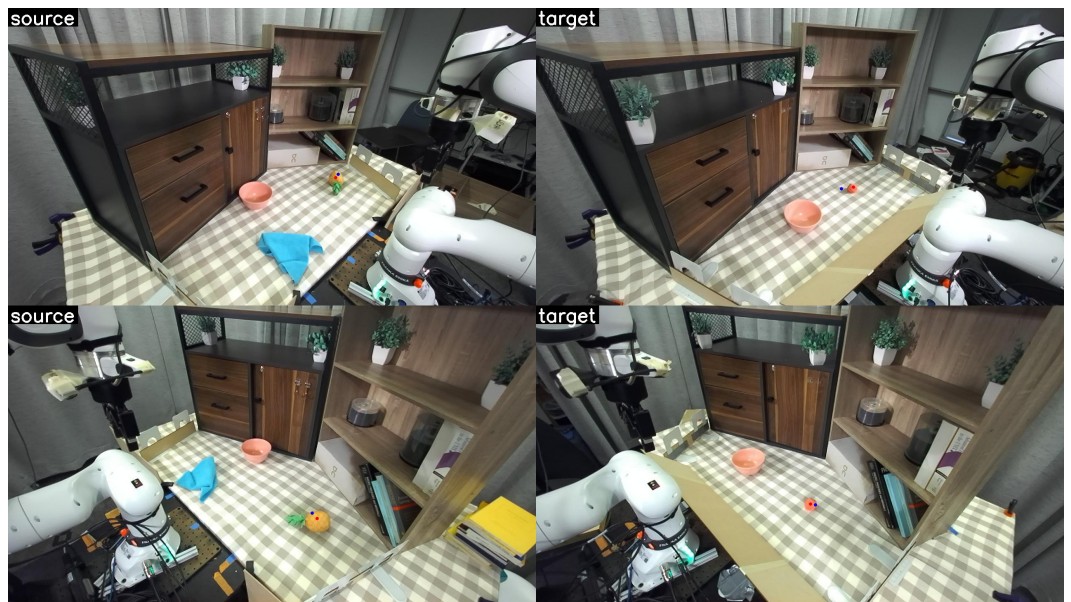

Figure 10: Computed correspondence for the Strawberry to Bowl task.

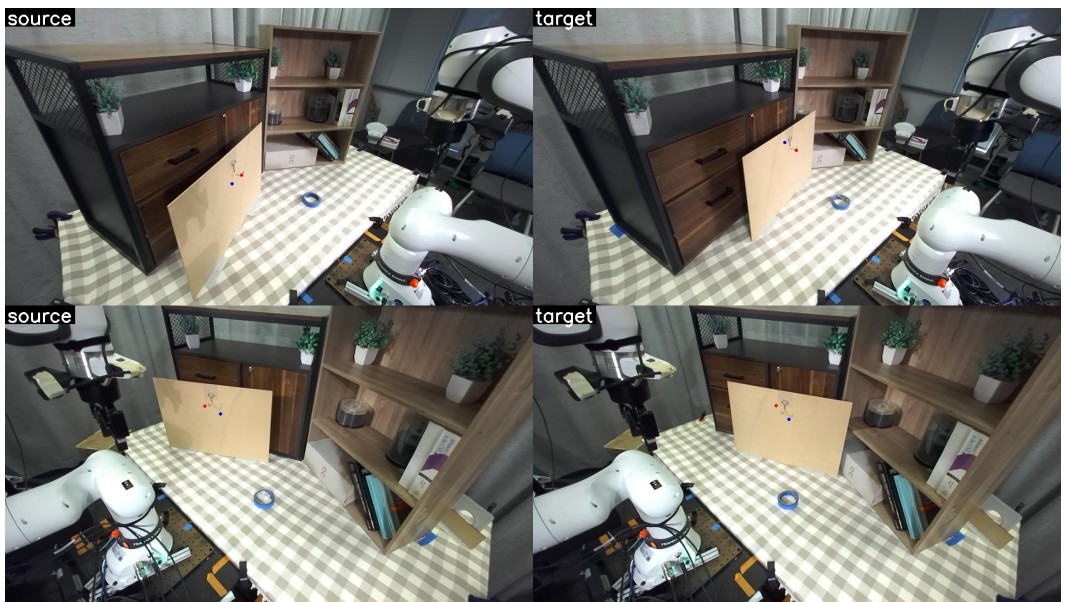

Figure 11: Computed correspondence for the Hanging Tape task.

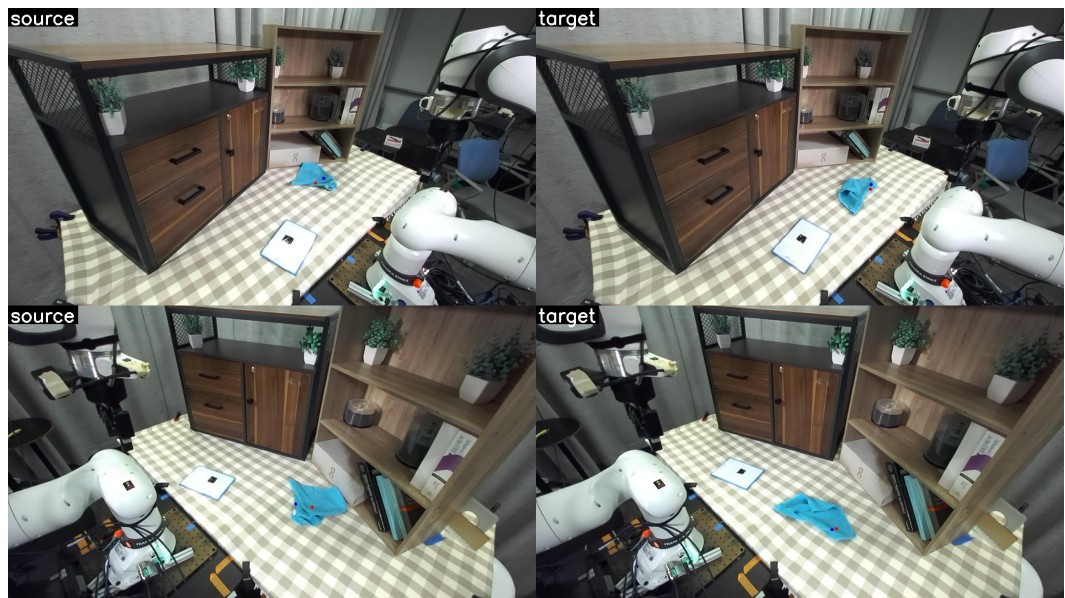

Figure 12: Computed correspondence for the Wiping with Cloth task.

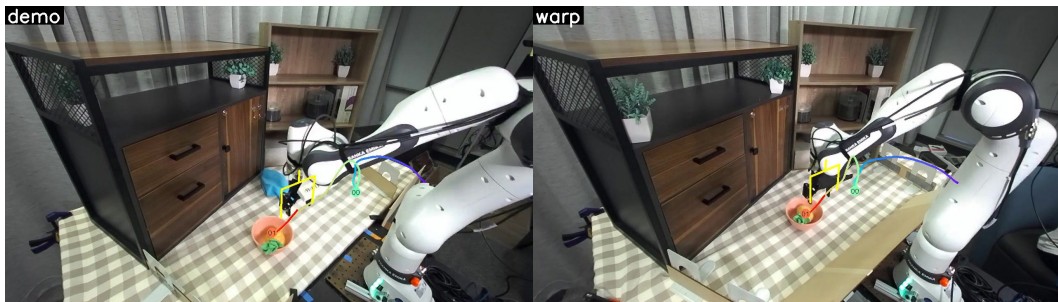

Figure 13: Trajectory warping for the Pineapple to Bowl task.

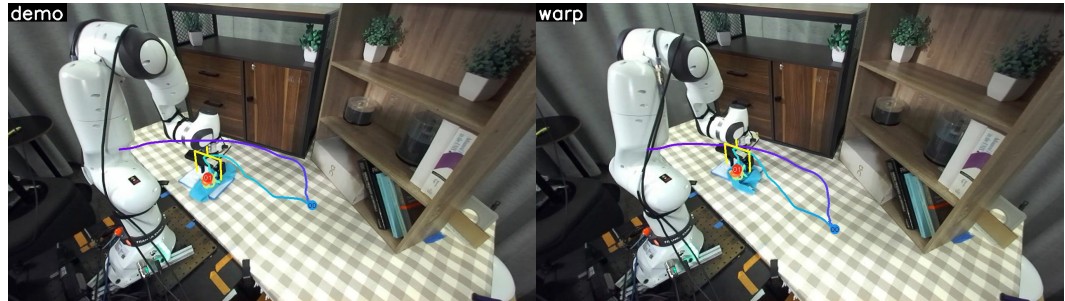

Figure 14: Trajectory warping for the Wiping with Cloth task.

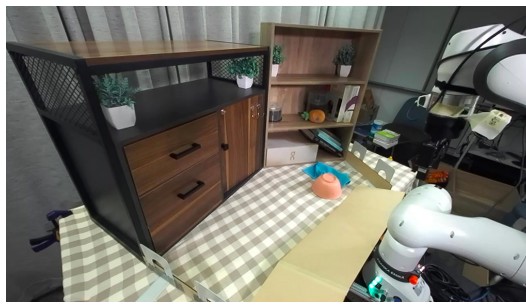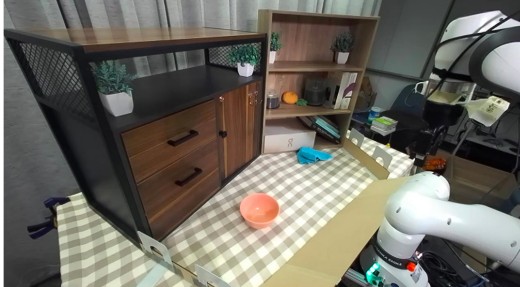

Figure 15: Environment states with the flipped bowl (left) or pineapple pushed into shelf (right) that required manual interventions.

| Method | Pineapple from Shelf | Pineapple to Bowl | Bowl to Shelf | Bowl from Shelf |
|---|---|---|---|---|
| Tether | $1.0 \pm 0.0$ | $1.0 \pm 0.0$ | $0.9 \pm 0.09$ | $1.0 \pm 0.0$ |
| KALM | $0.6 \pm 0.15$ | $0.0 \pm 0.0$ | $0.0 \pm 0.0$ | $0.5 \pm 0.16$ |

Table 2: **Policy Comparison with KALM.** Success rates and standard error measured over 10 trials.

**Play Timelapse.** We record a subsection of our 26-hour long autonomous play experiment. The timelapse, sped up by 100x, can be viewed on our website at `https://tether-research.github.io`.

**Play Interventions.** Among our 5 interventions, 4 involved the bowl flipping upside-down, and 1 involved the pineapple being pushed too far back into the shelf. While the latter situation occurs many times during play, this single instance is an edge case that could not be resolved autonomously via the Tether policy. Both situations are shown in Figure 15.

## A.5 EXTENDED EXPERIMENTAL RESULTS

Finally, we include additional experimental results regarding various aspects of our method.

**Additional Baselines.** To isolate and assess the importance of trajectory warping, we compare Tether with KALM (Fang et al., 2025), which also uses keypoint correspondences as input but produces actions via an imitation learning policy. Given 10 demos, our method outperforms KALM on our four pineapple-and-bowl tasks, as shown in Table 2. In general, we observe that KALM struggles with spatially generalizing to our large space of object poses, as its actions are predicted by an imitation learning policy, whereas Tether uses trajectory warping. Still, it does sometimes succeed at "Pineapple from Shelf" and "Bowl from Shelf," likely because the objects are randomly initialized on the smaller shelf; conversely, it fails to complete the other two tasks, where objects are randomized across the larger table. These results suggest that even with keypoint inputs, imitation learning falls behind trajectory warping in the low data regime, where a few demos usually cannot densely cover the initial object distribution and thus require the policy to have more structured generalization (e.g. via warping).

**Additional Environments and Tasks.** To evaluate the applicability and generalization of Tether, we run two additional experiments for "Pineapple to Bowl" in a mock kitchen setup and an office setup, with structural variation; in the former, we move the pineapple from the sink to a bowl on the counter, and in the latter, we move the pineapple from the top of a filing cabinet to a bowl inside a drawer. Our policy achieves 100% and 90% success respectively, thus validating that our method works well across different geometrically diverse environments, as well as spatial and semantic object variations.

Next, we assess our policy's potential to semantically generalize to unseen objects for more difficult tasks, beyond those reported in Figure 5 (row 2). Specifically, we run our policy for "Lifting the Handle" of a coffee machine: with 10 demos and evaluation on a new unseen machine, our policy achieves 80% success, thus demonstrating semantic generalization for a more complex task involving delicate articulation.

Finally, we evaluate our policy's ability to disambiguate between semantically similar objects: given 10 demos of "Pineapple to Bowl," "Strawberry to Bowl," and "Apple to Bowl," and evaluated in a

| Method | Pineapple from Shelf | Pineapple to Bowl | Bowl to Shelf |
|---|---|---|---|
| $\pi_0$ (10 human demos) | $0.0 \pm 0.0$ | $0.0 \pm 0.0$ | $0.0 \pm 0.0$ |
| $\pi_0$ (50 play-generated demos) | $0.1 \pm 0.09$ | $0.2 \pm 0.13$ | $0.1 \pm 0.09$ |
| $\pi_0$ (100 play-generated demos) | $0.3 \pm 0.14$ | $0.5 \pm 0.16$ | $0.1 \pm 0.09$ |
| $\pi_0$ (all play-generated demos) | $0.7 \pm 0.14$ | $0.8 \pm 0.13$ | $0.4 \pm 0.15$ |
| $\pi_0$ (pre-trained) | $0.5 \pm 0.16$ | $0.7 \pm 0.14$ | $0.0 \pm 0.0$ |

Table 3: **Downstream $\pi_0$ Finetuning Results.** Success rates and standard error measured over 10 trials.

multi-object setting with all three fruits present on the table, Tether achieves 100%, 80%, and 100% success rates respectively (with no failures caused by targeting the wrong object). Thus, we confirm that in cluttered multi-object environments, Tether can still accurately identify the correct object to manipulate.

**Downstream $\pi_0$ Finetuning.** Following a similar setting to Figure 7, we validate the effectiveness of our play-generated data in improving pretrained VLA policies like $\pi_0$. Results are reported in Table 3. In all three tasks, we observe consistent improvement as we include more data. We observe that while both baseline pre-trained $\pi_0$ and $\pi_0$ finetuned with few demos often miss the object and become stuck and idle, $\pi_0$ finetuned with more demos is able to correctly move toward the object, and in case of misses, sometimes even successfully retry. This validates that Tether-generated data is also useful for improving the performance of large pre-trained policies.

