# OpenReview forum: "Autonomous Functional Play with Correspondence-Driven Trajectory Warping"
_ICLR.cc/2026/Conference — ICLR 2026 Poster_

### Official Review · Reviewer_CA2Q · 2025-10-16

**Soundness:** 3
**Presentation:** 3
**Contribution:** 3
**Rating:** 8
**Confidence:** 2

**Summary:**

This paper presents Tether, a system for autonomous robotic play that learns from its own interaction data without human supervision. The method combines a non-parametric correspondence-driven trajectory warping policy—which warps demonstration trajectories to new scenes using semantic keypoint correspondences—with a VLM-guided self-improvement loop that autonomously selects, executes, and evaluates tasks. Starting from only a few human demonstrations, Tether repeatedly performs tasks, identifies successes using a combination of geometric and language-based evaluation, and accumulates new high-quality trajectories for imitation learning. Experiments across 12 real-world manipulation tasks on a Franka Panda arm show strong few-shot performance, robust generalization to unseen objects, and continuous improvement through autonomous play. The system autonomously collects over 1,000 expert-quality trajectories in 26 hours, enabling downstream neural policies to match or surpass those trained on human-collected data.

**Strengths:**

Tether introduces an elegant and practical approach to autonomous play that avoids the dependence on large-scale pretrained policies or foundation models. The key technical idea, trajectory warping from visual correspondences, is conceptually simple but powerful, enabling robust generalization from as few as 10 demonstrations. The geometric formulation preserves spatial precision while providing semantic flexibility, allowing the method to generalize to unseen objects and scene layouts.
The system-level design is well thought out: integrating VLM-based task selection and success evaluation into a continuous play loop demonstrates genuine autonomy rather than mere offline augmentation. The experiments are impressively thorough, spanning spatial, semantic, and dynamic challenges, and even covering deformable and articulated manipulation. The real-world deployment shows convincing scale and stability, with near-zero human intervention over long periods. Finally, the downstream imitation learning results clearly validate that the autonomously generated data is useful and progressively improves learned policies, providing a strong empirical story of self-improving robotics.

**Weaknesses:**

1. The trajectory warping mechanism makes a strong simplifying assumption of linear spatial interpolation between matched waypoints, ignoring the coupled effects of gripper orientation, contact geometry, and dynamic feasibility. This can distort end-effector poses for curved or articulated motions such as “Bowl to Shelf” or “Wiping with Cloth,” yet the paper provides no analysis or constraint mechanism to prevent physically invalid interpolations.

2. The correspondence-based matching assumes precise stereo geometry and reliable keypoint alignment, but the paper does not quantify how correspondence errors propagate to 3D waypoint accuracy. Since failed backprojections are simply discarded as “infeasible matches,” it is unclear how often this occurs or how sensitive performance is to visual conditions like lighting or occlusion.

3. The autonomous play loop relies heavily on VLM reasoning for both task sequencing and success evaluation, but the paper treats these modules as black boxes. There is no quantitative assessment of VLM reliability, error rates, or false success detections, which makes it difficult to judge how robust the overall autonomy pipeline truly is.

4. Lastly, while the downstream policy improvements are impressive, the analysis does not disentangle quantity from diversity of collected data. Without measuring coverage or novelty of the self-generated trajectories, it remains unclear whether improvement arises from meaningful exploration or simply repeated collection of similar trajectories.

**Questions:**

please address the concerns above

---

> ### Author Response · Authors · 2025-11-19
>
> Dear reviewer CA2Q, thank you for your thorough comments and feedback! Here, we respond to your specific questions and concerns.
>
> ---
>
> **Question/Comment 1**: The trajectory warping mechanism makes a strong simplifying assumption of linear spatial interpolation between matched waypoints, ignoring the coupled effects of gripper orientation, contact geometry, and dynamic feasibility. This can distort end-effector poses for curved or articulated motions such as “Bowl to Shelf” or “Wiping with Cloth,” yet the paper provides no analysis or constraint mechanism to prevent physically invalid interpolations.
>
> **Response**: Thank you for pointing out this potentially confusing aspect of our method; we hope to resolve it here. Our trajectory warping does not perform simple linear spatial interpolation between waypoints: interpolation is performed on the waypoint displacements, which is the delta applied to the original action sequence (Sec 3.1); this is carefully designed to preserve both trajectory shape and velocity, such as for “Bowl to Shelf” and “Wiping with Cloth.” Visualizations of the warped trajectories are in Appendix A.4, Fig 12 and 13.
>
> While we could push policy performance even further, we find that the current design is sufficient for its purpose of generating data through play, as it does not need to perfectly succeed every time. Nonetheless, future policy improvements is an important direction for future work, which may explore deploying an online residual policy on top of the warped action plan to improve contact geometry and dynamic feasibility.
>
> ---
>
> **Question/Comment 2**: The correspondence-based matching assumes precise stereo geometry and reliable keypoint alignment, but the paper does not quantify how correspondence errors propagate to 3D waypoint accuracy. Since failed backprojections are simply discarded as “infeasible matches,” it is unclear how often this occurs or how sensitive performance is to visual conditions like lighting or occlusion.
>
> **Response**: Indeed, as you mentioned, our method is robust against correspondence noise by filtering out failed backprojections. During autonomous play, we observe that ~50% of correspondences are filtered by this mechanism. We can afford this because (1) correspondence computation is efficient (0.05s per demo) and (2) we only need one successful correspondence out of many potential source demos.
>
> Regarding lighting and occlusion: we perform play across all hours of the day, and as our setup is near a window, the environment lighting does naturally shift over time; during this, we did not observe significant success rate changes. As for correspondences, we observe during play that Tether still works if the object is partially occluded (eg, pineapple inside the bowl), but correspondence will fail in the presence of heavy occlusion. Still, correspondence is of course inherently more susceptible to partial object occlusions than methods operating on full image observations, but this comes with large benefits for sample efficiency, and our experiments demonstrate that the resulting system is still robust enough to be useful.
>
> ---
>
> **Question/Comment 3**: The autonomous play loop relies heavily on VLM reasoning for both task sequencing and success evaluation, but the paper treats these modules as black boxes. There is no quantitative assessment of VLM reliability, error rates, or false success detections, which makes it difficult to judge how robust the overall autonomy pipeline truly is.
>
> **Response**: Thank you for suggesting these experiments, we have now performed evaluations of our VLM-based planning and evaluation. For task planning, we judge a VLM response as correct if its plan is reasonable and feasible, and for success detection, we compare the binary prediction with a human evaluation. Evaluated across all ~2000 play trajectories, task planning is 94.1% correct, and success detection achieves 97.7% precision (at 87.7% recall); as you mentioned, we prioritize precision as it’s most important to minimize false positives that pollute the success data.
>
> In addition, we perform the same tests with Gemini Robotics-ER 1.5, a new VLM trained specifically for robotic tasks like success evaluation and released after submission. It predicts 95.2% correct plans, and has 98.4% precision (at 89.6% recall) on success evaluation without the need for correspondence-based filtering. As Tether is agnostic to the choice of VLM, it can readily inherit these improvements.

---

> > ### Author Response · Authors · 2025-11-19
> >
> > **Question/Comment 4**: Lastly, while the downstream policy improvements are impressive, the analysis does not disentangle quantity from diversity of collected data. Without measuring coverage or novelty of the self-generated trajectories, it remains unclear whether improvement arises from meaningful exploration or simply repeated collection of similar trajectories.
> >
> > **Response**: Thank you, we do attempt to assess the diversity and quality of the collected data in two ways. First, we directly visualize coverage and novelty of successful trajectories by plotting a heatmap of their waypoints, which also serve as a proxy for object locations (Fig 6, right). We see that these waypoints evenly cover the table and shelf, and thus confirm that our generated data is indeed diverse and expands the distribution of our initial demo data.
> >
> > Second, we’d like to highlight that for the downstream policy improvements in Fig 7, the test trials are measured on a wide randomized initial state distribution, which is very poorly sampled by the initial few source demos. Thus, high performance naturally requires that the data generation be diverse, as it must densely cover the entire distribution rather than narrowly concentrating on the source demos. While evaluating policies from 0 to 2000 iterations, we qualitatively observe that the nature of their improvement lies in increased spatial robustness to different object positions; earlier policies succeed only for certain object locations, and final policies are generally effective for any random object placement.

---

> > > ### Author Response · Authors · 2025-11-24
> > >
> > > Dear reviewer CA2Q, thank you once again for your time and thoughtful review. We sincerely appreciate your feedback!
> > >
> > > We wanted to follow up and check if you have any remaining questions or concerns. We're happy to provide any further clarification or information if needed. Thank you!

---

### Official Review · Reviewer_x1Cz · 2025-10-28

**Soundness:** 2
**Presentation:** 3
**Contribution:** 2
**Rating:** 2
**Confidence:** 4

**Summary:**

This paper introduces **Tether**, a system that allows robots to learn from "autonomous play" instead of human-collected datasets. The Tether method consists of two primary components:

1. A non-parametric manipulation policy that operates using "correspondence-driven trajectory warping". Given only a few demonstrations, this policy finds semantic keypoint correspondences between a demo pre-image and the current scene. It then selects the best-matched demonstration and "warps" its 3D trajectory to fit the new scene.

2. An autonomous, multi-task "play" procedure that uses this policy to continuously generate new data by querying Vision-Language Models (VLMs) for high-level task selection, planning, and evaluation for success.

The authors demonstrate in real-world experiments that their policy, trained on only 10 demos, outperforms baselines like Diffusion Policy (DP) and VLA models ($\pi_0$). The policy shows generalization to OOD objects and spatial configurations and succeeds on deformable and high-precision tasks.
The autonomous play dataset can further train a Diffusion Policy, which progressively improves and achieves competitive performance.

**Strengths:**

1. **Effective Warping Policy**: Using semantic correspondences to directly warp a full trajectory is a data-efficient approach, outperforming the baselines.

2. **Strong Empirical Result**: The policy demonstrates generalization to OOD layouts and objects. The evaluation tasks include challenging ones involving high-precision and deformable objects.

3. **Comprehensive Evaluation**: The paper successfully closes the loop. It shows not only that the policy can generate data, but also that the generated data is useful by training a downstream Diffusion Policy, which achieves good performance.

4. **Clarity and Presentation**: The paper is well-written and easy to follow.

**Weaknesses:**

1. **Open-Loop Execution**: The policy appears to be entirely open-loop at the trajectory level. It computes correspondences, generates a full "action plan. Without a straightforward adjustment to closed-loop execution, this may create limitations on task complexity and horizon.

2. **Task Structure**: The authors claim the play method has "approximately indefinitely composable" structure. However, it is achieved on a deliberately designed, table-top task set. For tasks that are not easily invertible (e.g, washing dishes), the system may not be able to traverse a known, cyclic task graph.

3. **Baseline Comparisons**: The policy takes keypoints as inputs, which is mostly compared with baselines taking raw pixel images as inputs. Such comparisons could not effectively evaluate the performance of warping vs. imitation learning. Although the author includes KAT as one of the baselines, it is not effectively reproduced. Some simpler keypoint-based policies [1,2] could be included as baselines.

4. **Analysis of Failure Modes**: Given that the core idea of the paper is to make the robots play "autonomously", the analysis of the failure modes is more important, which is not covered in the paper.

**Questions:**

1. In line 182, how are the keypoints "identified"? Where are the keypoints in Figure 1? Moreover, the waypoints are determined by locations where the gripper toggles. This heuristic seems specific to pick-and-place. How does this definition apply to tasks like "Wiping with Cloth," where the critical part of the trajectory may not involve gripper state changes?

2. How robust is trajectory warping for articulated objects? If geometry ratios differ (e.g., cabinet openings that are higher/wider/deeper), does the warping generalize? Can it achieve cross-object generalization like [2]?

3. Human demonstrations often contain recovery from mistakes, creating data diversity in the strategy level, which trajectory warping may not capture. Is Tether primarily augmenting background/object pose? How does it compare with open-loop motion planning methods (e.g., RRT)?

4. What are the requirements for the task set? Specifically, what are the criteria to ensure cyclic task graph? How to determine if a task need to be segmented into mutiple subtask?

5. The policy finds the nearest-neighbor demo by computing correspondences against all demos in the set $\mathcal{D}$. How does the inference time of this policy scale as the number of demonstrations grows?

6. The core contribution of this paper seems to be the "autonomous data generation" instead of the policy. However, the paper mainly focus on the performace of the policy. How does its data generation compare to methods like [3,4]?

[1] Haldar, Siddhant, and Lerrel Pinto. "Point policy: Unifying observations and actions with key points for robot manipulation." CoRL, 2025.

[2] Fang, Xiaolin, et al. "KALM: Keypoint Abstraction Using Large Models for Object-Relative Imitation Learning." ICRA, 2025.

[3] Mandlekar, Ajay, et al. "Mimicgen: A data generation system for scalable robot learning using human demonstrations." CoRL, 2023.

[4] Pan, Chuer, et al. "One Demo is Worth a Thousand Trajectories: Action-View Augmentation for Visuomotor Policies." CoRL. 2025.

---

> ### Author Response · Authors · 2025-11-19
>
> Dear reviewer x1Cz, thank you for your thorough comments and feedback! Here, we respond to your specific questions and concerns.
>
> ---
>
> **Question/Comment 1**: Open-Loop Execution: The policy appears to be entirely open-loop at the trajectory level. It computes correspondences, generates a full "action plan.” Without a straightforward adjustment to closed-loop execution, this may create limitations on task complexity and horizon.
>
> **Response**: Tether aims to learn functional manipulation policies from a tiny number of demos, an order of magnitude smaller than most prevalent approaches today. This is only possible with strong inductive biases, and good design identifies and encodes biases that still permit handling a large space of tasks with reasonable proficiency.
>
> In our case, one of our core assumptions is that there isn’t any unexpected change in the environment during task execution, such as a person moving objects around after the robot begins to execute its plan. This is key to Tether’s efficacy in the very-low data regime: it needs only to learn actions for the initial state distribution, without needing to also handle intermediate states, and thus does not suffer from the classic compounding error issues of closed-loop methods.
>
> Every prior approach for few-shot policy learning [1, 2, 3] has had to make similarly strong assumptions, eg, open-loop executions, action primitives, keypoints as observations, etc. Still, the tasks we evaluate for Tether (Fig 4) are of similar difficulty with related work [1, 2, 3], even including those that use many more demos. Moreover, longer horizon tasks could be accomplished by chaining low-level Tether policies with high-level planning, following prior work [4, 5]—this is what we do during autonomous play (Sec 3.2).
>
> [1] Di Palo et al. Keypoint Action Tokens Enable In-Context Imitation Learning in Robotics
> [2] Myers et al. Policy Adaptation via Language Optimization: Decomposing Tasks for Few-Shot Imitation
> [3] Motion Tracks: A Unified Representation for Human-Robot Transfer in Few-Shot Imitation Learning
> [4] Ahn et al. Do As I Can, Not As I Say: Grounding Language in Robotic Affordances.
> [5] Huang et al. Inner Monologue: Embodied Reasoning through Planning with Language Models.
>
> ---
>
> **Question/Comment 2**: Task Structure: The authors claim the play method has "approximately indefinitely composable" structure. However, it is achieved on a deliberately designed, table-top task set. For tasks that are not easily invertible (e.g, washing dishes), the system may not be able to traverse a known, cyclic task graph… What are the requirements for the task set? Specifically, what are the criteria to ensure cyclic task graph? How to determine if a task needs to be segmented into multiple subtasks?
>
> **Response**: Indeed, as described in Sec 3.2, our task structure must be cyclic in order to avoid human resets and interventions; this is an extension of prior reset-free learning [1]. Our key insight is that there is a large space of useful tasks that satisfy this assumption, as the majority of tasks are invertible. With the exception of some cases (eg, breaking an object), most tasks like washing dishes can be inverted with eg, (1) a task that wipes the dishes and (2) a task that dispenses ketchup onto the dishes. In other words, we can consider that for most tasks that are teleoperated or evaluated in real [2, 3], with human researchers resetting the trials, we can construct our task structure by replacing the human reset with the equivalent robot task.
>
> The requirements are detailed in Sec 3.2 and Appendix A.2, which we summarize here as: for each task, the support of the final state distribution (including mistakes) is fully covered by the union of supports of the initial state distributions across all tasks. This means there is always a valid task to execute after a previous one, thus forming the cyclic structure.
>
> To determine if a task should be broken up into subtasks, we simply check whether it can be performed via multiple subsequent executions of Tether; it is always advantageous to break up such subtasks, as it can only expand the aforementioned “union of initial state supports” (by converting intermediate states to new subtask initial states) and thus improves the robustness of our cyclic structure.
>
> [1] Eysenbach et al. Leave no Trace: Learning to Reset for Safe and Autonomous Reinforcement Learning.
> [2] Chi et al. Universal Manipulation Interface: In-The-Wild Robot Teaching Without In-The-Wild Robots.
> [3] Physical Intelligence. π0: A Vision-Language-Action Flow Model for General Robot Control.

---

> ### Author Response · Authors · 2025-11-19
>
> **Question/Comment 3**: Baseline Comparisons: The policy takes keypoints as inputs, which is mostly compared with baselines taking raw pixel images as inputs. Such comparisons could not effectively evaluate the performance of warping vs. imitation learning. Although the author includes KAT as one of the baselines, it is not effectively reproduced. Some simpler keypoint-based policies [1,2] could be included as baselines.
>
> **Response**: We agree that additional baselines with keypoint-based policies are helpful for demonstrating the effectiveness of Tether; as [2] is most similar to our policy design, we are currently setting it up as an additional baseline experiment and will follow up with results soon. Additionally, we note that [1] is a contemporaneous work not yet published at time of submission, but we have looked into it and observe that while its policy also uses keypoints, its problem setting and assumptions are different: it is evaluated on a relatively narrow spatial distribution (ie, small variations in object pose), which differs from the diverse out-of-distribution scenarios our policy must handle during play.
>
> We would also like to note that we took every care to reproduce KAT based on the paper. We conducted an extensive set of ablations and simplifications, which are fully described in Appendix A.3. Unfortunately, in our settings, even when provided with ground-truth sensing (eg, human-labeled keypoints), multiple LLM models are still unable to produce an accurate trajectory (which we tested across multiple subsampling Hz). We hypothesize that this is because our trajectories, with orientation changes, non-linear velocities, and multi-object interactions, contain high-dimensional patterns too complex for a LLM to understand via in-context learning; in the original work, KAT was proven to work well on single object-centric tasks, rather than ones that span the entire workspace.
>
> ---
>
> **Question/Comment 4**: Analysis of Failure Modes: Given that the core idea of the paper is to make the robots play "autonomously", the analysis of the failure modes is more important, which is not covered in the paper.
>
> **Response**: As we only had 5 intervention-necessitating failures over 26 hours, the number of samples is too small to permit any comprehensive error analysis. Nevertheless, we discuss failure cases in the paper (Sec 4.3, Lines 421 to 451), of which the majority is caused by the bowl flipping completely upside down—a state that’s nearly impossible to fix with our single-arm embodiment, as un-flipping the bowl requires two separate points of contact. Interestingly, our method was able to recover from this state on two occasions by chance, when it accidentally pushed the bowl against the shelf and thus had two points of contact to flip it. This highlights the interesting nature of play: that at scale, coincidences result in unexpected novel behaviors.
>
> Other failure modes that we discovered and resolved in previous iterations of play experiments include crashing into the furniture, which we minimize via a simple OOB check, and dropping objects to the floor, which we avoid by placing barriers on the boundary of the robot’s workspace. We will expand this discussion in the revised draft and include more failure examples in the Appendix, thank you for the suggestion!

---

> ### Author Response · Authors · 2025-11-19
>
> **Question/Comment 5**: In line 182, how are the keypoints "identified"? Where are the keypoints in Figure 1? Moreover, the waypoints are determined by locations where the gripper toggles. This heuristic seems specific to pick-and-place. How does this definition apply to tasks like "Wiping with Cloth," where the critical part of the trajectory may not involve gripper state changes?
>
> **Response**: Keypoints are computed by projecting waypoints onto the image. In Figure 1, keypoints are at the same location as waypoints (as the visualization inherently projects the waypoints onto the image). We will make this clearer in the revision.
>
> Indeed, we choose waypoint extraction via gripper state change, which is commonly used in well-known prior work [1, 2], as stated in Sec 3.1. In our “Wiping with Cloth” task, we release the cloth next to the whiteboard, and thus implicitly define the wiping region as the area where the gripper opens.
>
> Nonetheless, our choice of waypoint extraction is orthogonal to the core method, and can be swapped out for alternatives. As an example, we experiment with “Pouring from Cup to Bowl,” which is difficult because (1) it requires a waypoint at the pouring location, which cannot be extracted via gripper change as the gripper is holding the cup, and (2) the keypoint used for corresponding the bowl does not lie at the gripper position, which is hovering above the bowl. In this case, we query a VLM to select keyframes from the demo video and point to the 2D pixel for the waypoint; then, we triangulate the pixels from both views to compute the waypoint. Finally, we continue the original algorithm to run correspondence and warp the trajectory. This method successfully finds waypoints on the cup and bowl, achieving 90% on pouring.
>
> [1] Shridhar et al. Perceiver-Actor: A Multi-Task Transformer for Robotic Manipulation.
> [2] James et al. Q-attention: Enabling Efficient Learning for Vision-based Robotic Manipulation.
>
> ---
>
> **Question/Comment 6**: How robust is trajectory warping for articulated objects? If geometry ratios differ (e.g., cabinet openings that are higher/wider/deeper), does the warping generalize? Can it achieve cross-object generalization like [2]?
>
> **Response**: Trajectory warping will be generally robust to articulated objects, given that their articulation geometry is similar (eg, both prismatic or revolute, and similar geometry ratios). However, warping will not generalize to vastly different articulations; this is because it’s challenging to infer articulation constraints from just the initial scene image. Still, we can partially circumvent this limitation by populating our demo set with a diverse variety of articulated objects. Then, by selecting the closest-matched demo, we will retrieve a demo performed on a similar object and thus be able to generalize.
>
> Note that the discussion above assumes sustained contact with the articulated object, as we show in “Opening Cabinet.” If this is not necessary, as with “Lifting the Handle” in [2], then the constraints are more relaxed. In either case, cross-object generalization is achievable due to semantic correspondence. To validate this, we perform the same experiment from [2]: with 10 demos lifting the handle of a coffee machine and evaluation on a new unseen coffee machine, Tether policy achieves 80% success.
>
> ---
>
> **Question/Comment 7**: Human demonstrations often contain recovery from mistakes, creating data diversity in the strategy level, which trajectory warping may not capture. Is Tether primarily augmenting background/object pose?
>
> **Response**: We’d like to first highlight that Tether does afford a kind of recovery: restarting execution after a failure will often correct the mistake. This is done in autonomous play (albeit with a potentially-different task for the subsequent execution), and this is why we only need 5 interventions in 26 hours. In fact, this form of recovery data possesses an advantage over human demos: humans usually recover immediately after making the mistake, so both recovery and mistake are in the same training trajectory and thus may require per-timestep filtering or weighing [1, 2] ; conversely, Tether already filters out its failed executions, so our generated data contains only the corrections, without mistakes.
>
> Additionally, Tether indeed augments the object position and pose, which we find is key to the downstream policy improvement in Fig 7 (and described in Sec 4.3). While we did not quantify policy recovery behaviors in that result, we do observe during trials that the downstream policies will retry the tasks, even if the bowl is tilted on its side, and succeed.
>
> [1] Peng et al. Advantage-Weighted Regression: Simple and Scalable Off-Policy Reinforcement Learning.
> [2] Zhang et al. SCIZOR: A Self-Supervised Approach to Data Curation for Large-Scale Imitation Learning.

---

> ### Author Response · Authors · 2025-11-19
>
> **Question/Comment 8**: How does Tether compare with open-loop motion planning methods (e.g., RRT)?
>
> **Response**: We are not sure we understand the request for comparison to open-loop motion planning approaches, in context of play data generation. Below, we compare motion planning to our few-shot policy design. Please let us know if we have addressed your question.
>
> Motion planning approaches like RRT typically require explicit models of the environment (geometry, dynamics, constraints), unlike Tether which simply learns from a small number of demonstrations. Even with this, they struggle to handle deformations and sustained contacts (eg, our “Wiping with Cloth” task), articulation (eg, our “Opening Cabinet” task), and precision (eg, our “Hanging Tape” and “Inserting Coffee” tasks). Generalization to unseen objects or layouts is hard, unlike Tether which flexibly handles semantic shifts like pineapple to apple or strawberry. Finally, motion planning approaches typically solve the problem anew for each scene configuration. This is quite computationally intensive due to collision checking, constraint solving, sampling-based search etc. unlike Tether which simply retrieves and warps previously seen successful trajectories.
>
> ---
>
> **Question/Comment 9**: The policy finds the nearest-neighbor demo by computing correspondences against all demos in the set . How does the inference time of this policy scale as the number of demonstrations grows?
>
> **Response**: Tether running on a single NVIDIA A6000 GPU takes 1.1 seconds of computation for feature extraction and 0.05s per demo for correspondence matching and nearest-neighbor selection, using under 24 GB GPU memory. Afterwards, warping to compute the action plan is very computationally light on CPU. Thus, while inference time scales with more demos, as with other non-parametric approaches, Tether runs quickly in the low-data regime; this is key to its high throughput during play, producing 1 success every 86 seconds (including trajectory execution time).
>
> ---
>
> **Question/Comment 10**: The core contribution of this paper seems to be the "autonomous data generation" instead of the policy. However, the paper mainly focus on the performance of the policy. How does its data generation compare to methods like [3,4]?
>
> **Response**: Indeed, the focus of our paper is autonomous data generation, which, as described in Sec 1, relies on a robust few-shot imitation policy; our Tether policy is well-suited for this, as it generalizes to diverse scene configurations (including OOD states caused by mistakes), and is data-efficient, requiring minimal human effort. To first verify these key properties, we evaluate Tether outside of the play procedure; Fig 5, row 1 is immediately relevant to play (by testing spatial variation), and rows 2 and 3 probe the limits of our policy design, attesting to its generality to other potential play tasks as well as its ability discover new tasks during play (eg, with unseen objects), as hinted by Reviewer QvvQ.
>
> Regarding comparing Tether with [3, 4]: our method is unique in that the augmented data is *real-world robot trajectories*, unlike MimicGen, which augments trajectories in simulation, or AVA, which generates synthetic trajectories with renderings of 3D reconstruction. Hence, these two alternatives have notable limitations:
> 1. MimicGen produces trajectories in simulation, and thus faces difficulties in sim-to-real policy transfer (reported in their Appendix) as RGB-based transfer for manipulation is challenging and remains an active area of research. Additionally, while they include two real-world results, their setup has substantial task-specific engineering (RANSAC, DBSCAN, pose estimation, ICP, etc) and requires manual resets.
> 2. AVA requires an accurate 3D reconstruction of the environment, and as their augmentation is synthetic and based on UMI, they note that some trajectories may not be kinematically feasible. Additionally, it was designed for the wrist camera, and thus cannot augment trajectories with external cameras, which are commonly used for policy training (eg, most VLA models).
>
> Another advantage of Tether is that its formulation was chosen to bridge augmentation with real-world learning [1, 2, 5], leaving it well-positioned for a key future direction: improving the data-generating policy over time.
>
> [1] Mendonca et al. Continuously Improving Mobile Manipulation with Autonomous Real-World RL.
> [2] Ahn et al. AutoRT: Embodied Foundation Models for Large Scale Orchestration of Robotic Agents.
> [5] Bousmalis et al. RoboCat: A Self-Improving Generalist Agent for Robotic Manipulation.

---

> > ### Author Response · Authors · 2025-11-24
> >
> > Dear reviewer x1Cz, thank you once again for your time and thoughtful review. We sincerely appreciate your feedback!
> >
> > We wanted to follow up and check if you have any remaining questions or concerns. We're happy to provide any further clarification or information if needed. We are also finishing follow-up experiments for Question/Comment 3, and expect to post results in the following few days. Thank you!

---

> > > ### Author Response · Authors · 2025-11-27
> > >
> > > Dear reviewer x1Cz, following up on our response to Question/Comment 3 (Baseline Comparisons), we have now run KALM [2] on our four pineapple-and-bowl tasks (Fig 5, row 1), with results below.
> > >
> > > | Method | Pineapple from Shelf | Pineapple to Bowl | Bowl to Shelf | Bowl from Shelf |
> > > |--------|------------------------|--------------------|----------------|------------------|
> > > | KALM | 60% | 0% | 0% | 50% |
> > > | Ours | 100% | 100% | 90% | 100% |
> > >
> > > In general, we observe that KALM struggles with spatially generalizing to our large space of object poses, as its actions are predicted by an imitation learning policy, whereas Tether uses trajectory warping. Still, it does sometimes succeed at “Pineapple from Shelf” and “Bowl from Shelf,” likely because the objects are randomly initialized on the smaller shelf; conversely, it fails to complete the other two tasks, where objects are randomized across the larger table. These results suggest that even with keypoint inputs, imitation learning falls behind trajectory warping in the low data regime, where a few demos usually cannot densely cover the initial object distribution and thus require the policy to have more structured generalization (eg, warping).
> > >
> > >
> > > As suggested in our previous response, we hypothesize that KALM and KAT were generally performant for tasks that have a more concentrated unimodal trajectory distribution centered around the target object (eg, robot always moves upward to “Lift Handle,” or orients sideways to “Pour”). In contrast, our tasks have a more diverse and uniform spatial distribution that covers the entire workspace (eg, for “Pineapple to Bowl,” the bowl can be anywhere relative to the pineapple); for such tasks, trajectory warping still works well because its spatial generalization is structurally defined by the warping procedure, instead of learned (via gradient descent, like KALM, or in-context, like KAT).
> > >
> > >
> > > To verify this hypothesis, we have run additional experiments with KALM and KAT on two tasks with narrower trajectory distributions: lifting the lid handle of a coffee machine, and opening a drawer. Our results, reported below, confirm that they indeed work well on these local object-centric interactions. This confirms that while our few-shot imitation baselines are performant on certain narrower trajectory distributions, Tether excels at handling more diverse, challenging trajectories.
> > >
> > > | Method | Lifting Handle | Opening Drawer |
> > > |--------|-----------------|----------------|
> > > | KALM | 50% | 80% |
> > > | KAT | 70% | 60% |
> > > | Ours | 90% | 100% |

---

### Official Review · Reviewer_QvvQ · 2025-10-30

**Soundness:** 2
**Presentation:** 2
**Contribution:** 3
**Rating:** 4
**Confidence:** 4

**Summary:**

This paper introduces Tether, a system for autonomous robotic manipulation that addresses the data scalability bottleneck in imitation learning through two main contributions: (1) a novel non-parametric policy that uses semantic visual correspondences to warp demonstrated trajectories into new scenes, achieving robust generalization across spatial and semantic variations with few demonstrations per task, and (2) a VLM-guided autonomous multi-task play procedure that enables the robot to continuously generate training data over extended periods (26 hours, producing 1000+ expert-level trajectories) with minimal human intervention. The key insight is that by leveraging strong visual priors from correspondence models rather than requiring massive demonstration datasets, the system can bootstrap from a few human demos to autonomously produce diverse, high-quality data that trains downstream neural policies to performance competitive with human-collected demonstrations, thereby shifting the scaling bottleneck from human time to robot time.

**Strengths:**

- The paper tackles an important problem in robot learning by demonstrating that autonomous play can generate over 1000 expert-level trajectories across 26 hours with minimal human intervention, offering a compelling alternative to labor-intensive human demonstration collection.
- The correspondence-driven trajectory warping policy demonstrates impressive robustness, outperforming strong baselines including vision-language-action models ($\pi_0$) and LLM-based methods (KAT).
- As evidenced in Fig.7, these tasks are reasonably complex that standard Diffusion Policy fails with limited data.

**Weaknesses:**

- While Fig.7 demonstrates that scaling to 2000 generated demonstrations improves Diffusion Policy performance, the paper lacks crucial comparisons with strong vision-language-action models like $\pi_0$, $\pi_{0.5}$ when provided with equivalent amounts of data. This comparison is critical because if VLAs can achieve high success rates with significantly fewer demonstrations (e.g., <500), it would diminish the practical value of scaling to 1000+ demonstrations per task.

- The current system generates demonstrations for tasks predefined by human experts, rather than discovering novel tasks beyond the initial demonstration scope. Works like BBSEA (arXiv:2402.08212) demonstrate that foundation models can autonomously propose new learnable tasks in unknown environments. While the present work successfully scales data collection for given tasks, extending the framework to autonomously expand the task repertoire (e.g., discovering new object interactions or compositional skills not present in initial demonstrations) would strengthen the contribution and better justify the "autonomous play" framing.

- The entire evaluation is conducted in one fixed environment configuration (table with two shelves and specific camera placements). While the method demonstrates robustness to spatial variations (object positions) and semantic variations (OOD objects) within this setup, testing across structurally different environments (e.g., different furniture arrangements, room layouts, or workspace configurations) would better validate the generalizability of the approach and strengthen claims about the method's broad applicability.

- The paper employs VLMs for task planning and success evaluation but provides no ablation studies on these critical components.

**Questions:**

- In the **Warping the Source Demo Trajectory** section, you describe interpolating 6-DOF gripper poses between waypoints, but it is unclear how the binary gripper open/close commands are handled during this warping process.

- For the **Out-of-Distribution Fruit and Containers** evaluation, is this transfer zero-shot (e.g., no strawberry demonstrations provided at all), or were OOD demos included in the given demonstrations?

**Details Of Ethics Concerns:**

No ethics review is needed.

---

> ### Author Response · Authors · 2025-11-19
>
> Dear reviewer QvvQ, thank you for your thorough comments and feedback! Here, we respond to your specific questions and concerns.
>
> ---
>
> **Question/Comment 1**: While Fig.7 demonstrates that scaling to 2000 generated demonstrations improves Diffusion Policy performance, the paper lacks crucial comparisons with strong vision-language-action models like 𝛑0,  when provided with equivalent amounts of data. This comparison is critical because if VLAs can achieve high success rates with significantly fewer demonstrations (e.g., <500), it would diminish the practical value of scaling to 1000+ demonstrations per task.
>
> **Response**: We’d like to first clarify that in 26 hours, our autonomous play procedure generates 1085 successful trajectories across all 6 tasks, or around 150-200 per task (Fig 6, left)—not >1000 per task. We find that this volume of data is sufficient to train the policies in Fig 7, which achieve high success rates from 80-100%. We will clarify these details in the revised draft.
>
> Nonetheless, we do agree that finetuning VLAs like pi_0 with the same play-generated data can further demonstrate the usefulness of our method, and we report results below. First, we see that pi_0 sees improvements in all 3 tasks, and thus our data is indeed helpful for VLA training. However, finetuned pi_0 performs worse than Diffusion Policy, which we hypothesize is because the latter model is smaller and specially trained for our environment and task, whereas finetuned pi_0 is much larger and may require more data for this specific task; this was also observed in prior work [1].
>
> In addition, we believe that the practical value of our method is actually complemented by methods like VLAs that scale well with more data: Tether can scale up demo datasets from very few demos, which in turn allows us to scale up model training (as in Fig 7). We further validate this by also finetuning pi_0 on the initial set of 10 provided demos, which causes overfitting and collapses the model. In contrast, by sending these initial demos to Tether and generating many more demos nearly autonomously, we can boost pi_0 performance instead.
>
> | Method | Pineapple from Shelf | Pineapple to Bowl | Bowl to Shelf |
> |----------------------------------|-----------------------|--------------------|----------------|
> | pi_0 | 50% | 70% | 0% |
> | Finetuned pi_0 (10 human demos) | 0% | 0% | 0% |
> | Finetuned pi_0 (>150 play demos) | 70% | 80% | 40% |
> | Diffusion Policy | 80% | 100% | 70% |
>
> [1] Kim et al. Fine-Tuning Vision-Language-Action Models: Optimizing Speed and Success.
>
> ---
>
> **Question/Comment 2**: The entire evaluation is conducted in one fixed environment configuration (table with two shelves and specific camera placements). While the method demonstrates robustness to spatial variations (object positions) and semantic variations (OOD objects) within this setup, testing across structurally different environments (e.g., different furniture arrangements, room layouts, or workspace configurations) would better validate the generalizability of the approach and strengthen claims about the method's broad applicability.
>
> **Response**: We agree that testing across different environments further showcases Tether’s applicability and generalizability, and have run two additional experiments for “Pineapple to Bowl” in a mock kitchen setup and an office setup, with structural variation; in the former, we move the pineapple from the sink to a bowl on the counter, and in the latter, we move the pineapple from the top of a filing cabinet to a bowl inside a drawer. The Tether policy achieves 100% and 90% success respectively. Thus, we validate that our method works well across different geometrically diverse environments, as well as spatial and semantic variations of objects.

---

> > ### Author Response · Authors · 2025-11-19
> >
> > **Question/Comment 3**: The paper employs VLMs for task planning and success evaluation but provides no ablation studies on these critical components.
> >
> > **Response**: Thank you for suggesting this experiment, we have now performed evaluations of our VLM-based planning and evaluation. For task planning, a human judges the VLM response as correct if its plan is reasonable and feasible (ie, current environment state satisfies the initial conditions for the plan), and for success detection, we compare the binary prediction with a human evaluation. Evaluated across all ~2000 play trajectories, task planning is 94.1% correct, and success detection achieves 97.7% precision (at 87.7% recall); note that we prioritize precision as it’s most important to minimize false positives that pollute the success data.
> >
> > We also ablate the need for VLM-based success evaluation by solely using our correspondence-based threshold. On its own, this mechanism achieves 85.2% precision (at 93.1% recall), with the majority of errors caused by noisy correspondences.
> >
> > Finally, we perform the same tests with Gemini Robotics-ER 1.5, a new VLM trained specifically for robotic tasks like success evaluation and released after submission. It predicts 95.2% correct plans, and has 98.4% precision (at 89.6% recall) on success evaluation without the need for correspondence-based filtering. As Tether is agnostic to the choice of VLM, it can readily inherit these improvements.
> >
> > Please let us know whether these experiments address your concerns, or whether you would suggest any other ablation studies.
> >
> > ---
> >
> > **Question/Comment 4**: The current system generates demonstrations for tasks predefined by human experts, rather than discovering novel tasks beyond the initial demonstration scope. Works like BBSEA (arXiv:2402.08212) demonstrate that foundation models can autonomously propose new learnable tasks in unknown environments. While the present work successfully scales data collection for given tasks, extending the framework to autonomously expand the task repertoire (e.g., discovering new object interactions or compositional skills not present in initial demonstrations) would strengthen the contribution and better justify the "autonomous play" framing.
> >
> > **Response**: We do indeed focus on generating new demos starting from a small initial set of demos, a problem of increasing importance that several recent efforts (Sec 2, Lines 100 to 125) have focused on. Perhaps a potential confusion here arises from the word “play,” which has often been used in an undirected exploration setting; to make our setting clearer, we will revise our framing to “autonomous task-directed play.”
> >
> > Proposing and gathering demos for new tasks is indeed interesting, but beyond the scope of this work. Prior approaches like BBSEA have taken advantage of ground-truth sensing, autonomous environment resets, and time acceleration in simulation to overcome the challenges here, whereas we instead focus on data generation in the real world. Nonetheless, we believe that Tether serves as a strong foundation for future work to readily tackle these challenges and thank the reviewer for this suggestion.
> >
> > ---
> >
> > **Question/Comment 5**: In the Warping the Source Demo Trajectory section, you describe interpolating 6-DOF gripper poses between waypoints, but it is unclear how the binary gripper open/close commands are handled during this warping process.
> >
> > **Response**: Since waypoints are defined via gripper open-close commands, the gripper does not change during the segments between waypoints. Thus, it is not affected when warping these segments; in the warped trajectory, gripper state change still occurs at the waypoints, just like the source trajectory, and remains constant in between.
> >
> > ---
> >
> > **Question/Comment 6**: For the Out-of-Distribution Fruit and Containers evaluation, is this transfer zero-shot (e.g., no strawberry demonstrations provided at all), or were OOD demos included in the given demonstrations?
> >
> > **Response**: The out-of-distribution experiments are performed entirely zero-shot—all demos are performed with the pineapple and pink bowl, and we visualize this in Appendix, Fig 8 and 9. Thus, we show that Tether can semantically generalize to unseen objects, and has high potential for future work such as novel task discovery, as described in Question/Comment 4.

---

> > > ### Author Response · Authors · 2025-11-24
> > >
> > > Dear reviewer QvvQ, thank you once again for your time and thoughtful review. We sincerely appreciate your feedback!
> > >
> > > We wanted to follow up and check if you have any remaining questions or concerns. We're happy to provide any further clarification or information if needed. Thank you!

---

> ### Comment · Reviewer_QvvQ · 2025-11-25
>
> Thank you for your response. Here are my follow-up comments:
>
> **Regarding Q1**, I believe you should provide more fine-grained ablation studies on VLA fine-tuning data requirements. While 10 demonstrations per task proved insufficient, it remains unclear whether the full 150+ demonstrations are necessary. The gap between 10 and 150 is substantial, and understanding the performance curve in this range is important for assessing practical applicability. Could you please:
> - Provide results with intermediate fine-tuning data amounts (e.g., 50, 100 demos) to characterize the data efficiency curve?
> - Include video rollouts of all policy results in the supplementary materials to facilitate qualitative comparison across different data regimes?
>
> **Regarding Q6**, I have a significant concern about this aspect of your approach. In scenarios with multiple objects of the same category (e.g., multiple graspable objects in a scene), it is unclear how your policy determines which specific object to interact with, especially since your method appears to provide no explicit disambiguation signal (e.g. you still provide a demonstration with a pineapple instead of a strawberry). Could you please:
> - Clarify the mechanism by which your policy handles multi-object scenarios?
> - Explain what information disambiguates the target object when multiple valid candidates exist?
> - Provide experimental results or qualitative examples demonstrating robust behavior in multi-object settings?
> - If disambiguation is currently not handled, discuss how this limitation affects the claimed capabilities?
>
> This is a critical issue for the policy component of your work. Without a clear explanation and experimental validation of multi-object handling, I may need to revise my assessment of the policy results downward. Please address this carefully in your revision, as it directly impacts the practical applicability and robustness of your approach.

---

> ### Author Response · Authors · 2025-11-27
>
> Dear reviewer QvvQ,
>
> Thank you again for your time and for engaging in the discussion period to allow us to better address your concerns.
>
> ---
>
> **Re Question/Comment 1**: Please recall that the objective of these experiments is to show that the **demonstrations collected by our autonomous play system are useful to train reactive policies**. We had first demonstrated these with diffusion policies trained from scratch in the submission, and upon your request, by finetuning pi_0 VLA policies. In particular, we have shown both that:
> 1. Pretrained VLAs do not improve through directly finetuning on the same 10 seed demos that our autonomous data generation setup requires.
> 2. Pretrained VLAs do indeed benefit from the expanded set of demos produced by our approach. Since VLA finetuning runs and robot evaluations are expensive to run, we picked the expanded dataset size (>150) to be representative of prior attempts to demonstrate VLA finetuning [1, 2], and to match our final DP experiments from Fig 7.
>
> Additionally, note that there are many good reasons to want to scale data beyond just their application for VLA finetuning. Today’s robotics datasets are tiny compared to foundation model training data in language and vision [3, 4, 5], and the question of scaling data for VLAs and other robotics foundation models (eg, world models and reward models) remains open. In this context, methods like ours that require minimal human intervention to scale real-world robot trajectories could be particularly valuable. In other words, **rather than devalue our approach, the data hungriness of robotics foundation model training actually emphasizes the importance of autonomous data generation**.
>
> Our argument above has been that studying the scaling laws of VLA finetuning with our demos is not critical to evaluating our approach. However, to answer your question and also out of our own curiosity about this, we have run the suggested pi_0 finetuning experiments with 50, 100, and >150 play-generated demos (162, 184, and 209 respectively, as reported in Fig 6). The performance curve is below. To better illustrate policy performance, we also include intermediate task success (picking up the correct object) in parentheses; videos of these trials are also included in supplemental materials.
>
> | Method | Pineapple from Shelf | Pineapple to Bowl | Bowl to Shelf |
> |-----------------------------------|-----------------------|--------------------|----------------|
> | pi_0 | 50% (50%) | 70% (70%) | 0% (70%) |
> | Finetuned pi_0 (10 human demos) | 0% (0%) | 0% (50%) | 0% (40%) |
> | Finetuned pi_0 (50 play demos) | 10% (10%) | 20% (20%) | 10% (20%) |
> | Finetuned pi_0 (100 play demos) | 30% (30%) | 50% (50%) | 10% (40%) |
> | Finetuned pi_0 (>150 play demos) | 70% (70%) | 80% (80%) | 40% (100%) |
>
> In summary, both baseline pre-trained pi_0 and pi_0 finetuned with few demos often miss the object and become stuck and idle; conversely, pi_0 finetuned with more demos is able to correctly move toward the object, and in case of misses, sometimes even successfully retry. Similar behaviors were found in recent studies on training large policies [6]. In addition, we see that all three tasks follow a clear scaling trend: finetuned pi_0 continues to benefit from additional data, and since performance on all three tasks has yet to saturate, it’s likely that **larger datasets are required to reach maximum finetuning performance, which further validates the value of our approach for scaling autonomous data generation**.
>
>
> [1] Kim et al. Fine-Tuning Vision-Language-Action Models: Optimizing Speed and Success.
> [2] Physical Intelligence. π0: A Vision-Language-Action Flow Model for General Robot Control.
> [3] Black. From Octo to π0: How to Train Your Generalist Robot Policy.
> [4] Goldberg. Good old-­ fashioned engineering can close the 100,000-­ year “data gap” in robotics.
> [5] Hu et al. Toward General-Purpose Robots via Foundation Models: A Survey and Meta-Analysis.
> [6] TRI LBM Team. A Careful Examination of Large Behavior Models for Multitask Dexterous Manipulation.

---

> > ### Author Response · Authors · 2025-11-27
> >
> > **Re Question/Comment 6**: Recall that the power of Tether to identify and handle relevant objects even in unseen scenes comes from visual correspondence matching (Sec 3.1), which has made large capability leaps in recent years. As we show in Sec 4.2 and Fig 5, this approach easily handles seen objects in new configurations even in cluttered multi-object scenes.
> >
> > For unseen objects, correspondence finding is still capable of finding the best semantic match to the training demo keypoints, as we explain in Sec 4.2: “without the demo object being present at test-time, correspondence finds the most semantically similar object” (eg, switching pineapple for apple, as we show in Fig 5, row 2). In other words, if a demo keypoint for “Pineapple to Bowl” were centered on the pineapple, and the target scene has an apple and the bowl (both of which are graspable objects), then correspondence matching successfully finds the apple. This ability fundamentally arises from the semantic visual embeddings exploited in our correspondence matching approach. Note also that our experiments on this aspect follow notable prior work [1, 2, 3, 4, 5] on handling unseen objects: like them, we switch out the demo object with a new unseen object (visualized in Appendix A.4, Fig 8 and 9).
> >
> > If the target scene has multiple plausible matches, such as multiple fruits (eg, two pineapples, or an unseen apple and an unseen strawberry), correspondence matching still selects one of them. Note that the specific way the tie is broken here is unimportant, since selecting any of those fruits would lead to an equally valid interpretation of how to transfer the training demos from the pineapple. In other words, this is about the best that one could be expected to do in this scenario, since the task in this out-of-distribution transfer setting is under-specified.
> >
> > Finally, if one wanted to avoid such under-specified settings and disambiguate semantically similar unseen objects (eg, apple and strawberry), one would simply provide demos targeting each object. This requires minimal effort as Tether uses very few demos, which can be collected in a few minutes per task. We validate that Tether does indeed work well in such cases: given 10 demos of “Pineapple to Bowl,” “Strawberry to Bowl,” and “Apple to Bowl,” and evaluated in a multi-object setting with all three fruits present on the table, Tether achieves 100%, 80%, and 100% success rates respectively (with no failures caused by targeting the wrong object).
> >
> > We will make all this more clear in the revised draft. Please let us know if this addresses your concern, and we are happy to provide further clarification if needed!
> >
> > [1] Fang et al. Keypoint Abstraction using Large Models for Object-Relative Imitation Learning.
> > [2] Di Palo et al. DINOBot: Robot Manipulation via Retrieval and Alignment with Vision Foundation Models.
> > [3] Vosylius et al. Instant Policy: In-Context Imitation Learning via Graph Diffusion.
> > [4] Di Palo et al. Keypoint Action Tokens Enable In-Context Imitation Learning in Robotics.
> > [5] Wen et al. You Only Demonstrate Once: Category-Level Manipulation from Single Visual Demonstration.

---

### Official Review · Reviewer_ZDhN · 2025-10-30

**Soundness:** 3
**Presentation:** 3
**Contribution:** 3
**Rating:** 4
**Confidence:** 3

**Summary:**

This paper presents Tether, a framework for autonomous robotic play that aims to reduce dependence on large-scale human demonstrations for imitation learning. The method combines a non-parametric, correspondence-driven policy for robust generalization with a vision-language model (VLM)-guided autonomous play loop that continuously generates new training data.

The core insight is to leverage semantic image correspondences between demonstrations and novel scenes to warp existing robot trajectories. This approach enables a robot to adapt to a small set of demonstrations (as few as 10) in new environments and tasks without requiring retraining. The authors then deploy this policy in a self-improving cycle of multi-task play: the robot autonomously selects, executes, and evaluates tasks using VLM reasoning for both planning and success detection.

Over 26 hours of autonomous operation, Tether produces over 1,000 successful real-world trajectories with negligible human intervention (~0.26% of runs requiring resets). The collected data are subsequently used to train downstream neural policies, which achieve success rates comparable to or better than policies trained on human demonstrations.

**Strengths:**

- The paper introduces a clever and practical way to reuse a small number of demonstrations via keypoint correspondence-driven trajectory warping. This method leverages advances in visual correspondence (e.g., DINOv2 + Stable Diffusion) to enable generalization across large spatial and semantic variations without retraining.

- Experiments on 12 diverse manipulation tasks—including deformable, articulated, and precision tasks—show consistent improvement over baselines such as Diffusion Policy, ε₀ (OpenVLA), and KAT. The approach exhibits impressive semantic generalization, successfully adapting demonstrations from, e.g., a pineapple to an apple or a bowl to a cup

**Weaknesses:**

- One major contribution, at the same time, can be a limitation as well, is the usage of correspondence. While correspondence-based warping enables generalization, its performance is tied to the quality and stability of keypoint matching. In cases with heavy occlusion, textureless surfaces, or large deformations, this component may degrade. It would be nice for the authors to include such discussions in the paper on which external disturbances the policy is robust to, and which disturbances it cannot handle.

- Non-parametric vs. generalization claim: The paper argues that a non-parametric policy generalizes to unseen objects and layouts. However, this non-parametric methods always require access to demonstrations at inference time, which limits scalability and contradicts the claim of “extreme generalization”. It would be helpful for the authors to explain this better in the text.

- Clarification question: How does the relatively coarse 10 cm correspondence threshold and the potential variability of VLM-based verification affect the reliability of results, especially for precision-sensitive tasks like coffee pod insertion with an 8 mm margin?

**Questions:**

See the points above. I am eager to see the paper get improved.

---

> ### Author Response · Authors · 2025-11-19
>
> Dear reviewer ZDhN, thank you for your thorough comments and feedback! Here, we respond to your specific questions and concerns.
>
> ---
>
> **Question/Comment 1**: One major contribution, at the same time, can be a limitation as well, is the usage of correspondence. While correspondence-based warping enables generalization, its performance is tied to the quality and stability of keypoint matching. In cases with heavy occlusion, textureless surfaces, or large deformations, this component may degrade. It would be nice for the authors to include such discussions in the paper on which external disturbances the policy is robust to, and which disturbances it cannot handle.
>
> **Response**: Thank you for suggesting these considerations, such a discussion would certainly improve the paper! We will include the following points in our revision.
>
> Indeed, learning from limited data requires strong built-in priors. Thus, our strength in exploiting semantic correspondences is also a weakness. We assume both that this abstraction of the scene into keypoints is sufficient, and that the abstraction can be reliably computed.
>
> For computing the abstraction, our method directly inherits the recent advances in correspondences that have occurred in computer vision. While feature tracking on textureless or deformable regions has historically been very difficult, modern approaches implicitly exploit global information to overcome such difficulties, and have had success in such settings [1, 2, 3, 4, 5]. We use one such state-of-the-art approach [5] for Tether, and observe that our correspondences work well for objects like the pink bowl (will add visualization to Appendix), which lacks texture, and the blue cloth (Appendix A.4, Fig 11), which is fully deformable.
>
> We observe during play that Tether still works if the object is partially occluded (eg, pineapple inside the bowl), but correspondence will fail in the presence of heavy occlusion. Still, correspondence is of course inherently more susceptible to partial object occlusions than methods operating on full image observations, but this comes with large benefits for sample efficiency, and our experiments demonstrate that the resulting system is still robust enough to be useful.
>
> [1] Oquab et al. DINOv2: Learning Robust Visual Features without Supervision.
> [2] Luo et al. Diffusion Hyperfeatures: Searching Through Time and Space for Semantic Correspondence.
> [3] Hedlin et al. Unsupervised Semantic Correspondence Using Stable Diffusion.
> [4] Tang et al. Emergent Correspondence from Image Diffusion.
> [5] Zhang et al. Telling Left from Right: Identifying Geometry-Aware Semantic Correspondence.
>
> ---
>
> **Question/Comment 2**: Non-parametric vs. generalization claim: The paper argues that a non-parametric policy generalizes to unseen objects and layouts. However, this non-parametric methods always require access to demonstrations at inference time, which limits scalability and contradicts the claim of “extreme generalization”. It would be helpful for the authors to explain this better in the text.
>
> **Response**: In Sec 4.2 of the paper, we claimed that Tether “excels at both spatial and semantic generalization,” which we demonstrate via experiments in Fig 5. This capability is inextricably tied with the few-shot learning and autonomous play setting that we target: from only ~10 demos, we intend to construct policies that can generalize to many new unseen configurations of the scene. Our insight is that this form of generalization can arise in the low-data regime by leveraging correspondence, based on vision foundation models, as a built-in structural prior; this is what allows our non-parametric policy to generalize to new object poses, as well as unseen objects.
>
> Note that this form of generalization is fundamentally different from that achieved by scalable parametric policy designs (eg, VLAs), which learn to generalize by training on thousands of diverse demos. With such large training dataset sizes, non-parametric approaches like Tether do indeed scale poorly (in terms of inference-time computation). However, at such dataset sizes, one does not require few-shot imitation or structured generalization, and Tether is therefore not well-motivated as a policy-learning approach for such settings.
>
> Thank you for pointing this out, we will expand on our explanation in the revised draft, as well as revise the “extreme generalization” phrase in our abstract to be consistent with the above wording. Please also let us know if this has addressed your concerns.

---

> > ### Author Response · Authors · 2025-11-19
> >
> > **Question/Comment 3**: Clarification question: How does the relatively coarse 10 cm correspondence threshold and the potential variability of VLM-based verification affect the reliability of results, especially for precision-sensitive tasks like coffee pod insertion with an 8 mm margin?
> >
> > **Response**: Thank you for pointing out this potentially confusing part of our experiment! We’d like to clarify that the 10 cm correspondence threshold is used in success detection during play, which involves the pineapple and bowl. The other tasks from Fig 5, like “Inserting Coffee,” were run solely for policy evaluations and are not part of play. All few-shot imitation results in Fig 5, as well as downstream policy results in Fig 7, had success evaluated by humans.
> >
> > Nevertheless, we have also now performed evaluations on the reliability of our correspondence threshold and VLM-based verification. Across all ~2000 play trajectories, our method achieves 97.7% precision (at 87.7% recall); note that we prioritize precision as it is most important to minimize false positives that pollute the success data.
> >
> > Additionally, since the time of submission, we have found that a correspondence-based threshold may be unnecessary: using only Gemini Robotics-ER 1.5, which was trained for robotic tasks like success evaluation and released after submission, we were able to categorize our play data with 98.4% precision (at 89.6% recall), as well as “Inserting Coffee” trials with 100% precision (at 75% recall). Once we have repeated all our experiments with this system, we can remove the correspondence threshold-based filtering component of our system to simplify it.

---

> > > ### Author Response · Authors · 2025-11-21
> > >
> > > Dear Reviewer ZDhN, thank you once again for your time and thoughtful review, and we sincerely appreciate your feedback.
> > >
> > > We believe we have addressed your main concerns, but please do let us know if any remain so that we may try to address them in the rest of the window; we’re happy to provide any further clarification or information if needed. Thank you!

---

> > ### Comment · Reviewer_ZDhN · 2025-11-24
> >
> > Thank you for the detailed response. I appreciate the explanation, the planned revisions, and the clarifications to my concerns. While occlusion remains a minor limitation, the overall contribution is clear, and the work provides meaningful insights for the community. I will raise my score accordingly.

---

### Official Review · Reviewer_3dVV · 2025-10-31

**Soundness:** 3
**Presentation:** 3
**Contribution:** 3
**Rating:** 8
**Confidence:** 3

**Summary:**

The paper proposed 1) a non-parametric table-top manipulation policy by warping source demonstrations, which has strong generalization performance, and 2) a framework for autonomously generating robot demonstrations with minimal human interventions. The proposed method is evaluated on 12 table-top manipulation tasks. The first part of the experiments shows the proposed non-parametric policy outperforms three baselines. The second part of the experiments shows the proposed framework can continuously gather demonstrations with minimal human interventions, and the downstream policy performance scales well with the autonomously-collected demonstrations.

**Strengths:**

- The paper addresses an important problem - scaling data collection. Through the experiments, the paper shows that it enables collection of a large number of robot demonstrations with minimal human interventions, and the collected demonstrations are of comparable quality to human demonstrations in terms of downstream policy learning performance.
- The paper is clear.

**Weaknesses:**

- The baselines for Section 4.2 are not fair comparison, and thus weakens the claim that the proposed Tether policy has superior generalization. Concretely, Tether is given 10 demonstrations, while $\pi_0$ with FAST tokenizer is evaluated zero-shot. Diffusion policy, not meant for few-shot setting, is also disadvantaged by being trained from scratch with 10 demonstrations.

**Questions:**

- The autonomous play framework proposed in Section 3.2 appears to be policy-agnostic. Can we use the same framework with different policies?

---

> ### Author Response · Authors · 2025-11-19
>
> Dear reviewer 3dVV, thank you for your thorough comments and feedback! Here, we respond to your specific questions and concerns.
>
> ---
>
> **Question/Comment 1**: The baselines for Section 4.2 are not fair comparison, and thus weakens the claim that the proposed Tether policy has superior generalization. Concretely, Tether is given 10 demonstrations, while pi_0 with FAST tokenizer is evaluated zero-shot. Diffusion policy, not meant for few-shot setting, is also disadvantaged by being trained from scratch with 10 demonstrations.
>
> **Response**: Thank you for pointing out this potentially confusing aspect of our baseline experiments; we hope to resolve it here. We chose our baselines to span the spectrum of expected data efficiency among widely prevalent approaches today: Diffusion Policy typically requires a few hundred demos, KAT (like Tether) is designed to work with only a few demos, and pi_0 is designed to run zero-shot. Together, these comparisons, besides confirming our prior expectations in data efficiency, also validate that (1) our low-data regime cannot be tackled by general-purpose methods like Diffusion Policy, (2) our Tether policy surpasses alternative data-efficient methods, and (3) our tasks cannot be simply achieved zero-shot by modern VLAs.
>
> Still, we agree that to provide a more fair comparison between the Tether policy and pi_0, we can finetune it (using their provided procedure) with the same 10 demos as we used to set up Tether. We have now run this experiment. However,  we find that with so few demos, pi_0 performance actually degrades after finetuning, potentially due to severe overfitting; this matches prior work on finetuning VLAs, which require 100s of demos [1, 2]. To validate this, we also finetune pi_0 using the play data collected by our Tether-based autonomous play system (>150 demos per task), and we observe consistent improvements. Results are in the table below. Besides validating our pi_0 and pi_0 finetuning comparisons, this also serves to further strengthen our claim that our autonomous data collection yields high-quality demonstrations useful for imitation learning.
>
> | Method | Pineapple from Shelf | Pineapple to Bowl | Bowl to Shelf |
> |----------------------------------|-----------------------|--------------------|----------------|
> | Ours | 100% | 100% | 90% |
> | pi_0 | 50% | 70% | 0% |
> | Finetuned pi_0 (10 human demos) | 0% | 0% | 0% |
> | Finetuned pi_0 (>150 play demos) | 70% | 80% | 40% |
>
> [1] Kim et al. Fine-Tuning Vision-Language-Action Models: Optimizing Speed and Success.
> [2] Physical Intelligence. π0: A Vision-Language-Action Flow Model for General Robot Control.
>
> ---
>
> **Question/Comment 2**: The autonomous play framework proposed in Section 3.2 appears to be policy-agnostic. Can we use the same framework with different policies?
>
> **Response**: Yes, we can indeed use the same framework with different policies! However, an effective policy must (1) be able to generalize spatially to diverse scene configurations, especially OOD states that can occur due to mistakes, and (2) be data-efficient, in order to start our process on new tasks with minimal human effort. Our trajectory warping policy is designed specifically for this purpose, and is very effective at achieving both key properties, as shown in our imitation learning experiments (Sec 4.2). We motivate this in Sec 3 in the submission, and we will strive to make this connection clearer in revisions.

---

> > ### Author Response · Authors · 2025-11-24
> >
> > Dear reviewer 3dVV, thank you once again for your time and thoughtful review. We sincerely appreciate your feedback!
> >
> > We wanted to follow up and check if you have any remaining questions or concerns. We're happy to provide any further clarification or information if needed. Thank you!

---

> ### Comment · Reviewer_3dVV · 2025-11-25
> **Post-rebuttal Comment**
>
> No further comment.

---

### Author Response · Authors · 2025-11-19

Dear AC and reviewers,

Thank you for reviewing and improving our work! We are glad that reviewers recognize our key contributions:
1. A **novel policy design** that excels in the **very-low data regime** (<=10 demos) across 12 comprehensive real-world tasks [3dVV, ZDhN, QvvQ, x1Cz, CA2Q].
2. **Autonomous play** for data generation, which to our knowledge is the first to use trajectory warping to produce **>1000 real-world demos for policy learning** [3dVV, QvvQ, x1Cz, CA2Q]

We’d like to highlight key discussions and new experiments surrounding primary concerns:
1. **VLM planning and success detection.** Our play procedure employs VLMs for planning and success detection. To verify the reliability of the VLM responses, we evaluate its responses across all ~2000 play trajectories and find that our task planning is 94.1% correct, and success detection achieves 97.7% precision (at 87.7% recall); note that we prioritize precision as it’s most important to minimize false positives that pollute the success data. Thus, VLM-based planning and success evaluation is largely robust for our play procedure. [[ZDhN Q3](https://openreview.net/forum?id=FqDmvMZish&noteId=TLTs4v6sJS), [QvvQ Q3](https://openreview.net/forum?id=FqDmvMZish&noteId=VDRnIKvfQF), [CA2Q Q3](https://openreview.net/forum?id=FqDmvMZish&noteId=ESBJul5hAt)]
2. **Finetuned VLAs.** Our original experiments compared Tether’s policy (given 10 demos) with pretrained pi_0. To provide a more fair comparison, we additionally finetune pi_0 with the same 10 demos; however, it struggles in this data regime, potentially due to severe overfitting. Alternatively, finetuning pi_0 with our Tether-generated data (50, 100, and >150 demos) results in performance gains and a consistent improvement curve as more demos are added. Altogether, this shows (1) Tether outperforms modern VLAs in the low data regime and (2) Tether is valuable for autonomously scaling real-world data for improving VLA policies. [[3dVV Q1](https://openreview.net/forum?id=FqDmvMZish&noteId=R0jaOJ1MGk), [QvvQ Q1](https://openreview.net/forum?id=FqDmvMZish&noteId=OzYeT5u5pC)].
3. **Keypoint-based baseline.** Our original experiments compared Tether with KAT [1], a prior keypoint-based few-shot imitation method, and found that it struggles in our tasks. To provide an additional comparison with another keypoint-based policy method, we set up and run the suggested KALM [2] for our four pineapple-and-bowl tasks. It falls behind Tether in all tasks, which demonstrates that trajectory warping is superior to KALM’s imitation learning in the low data regime; this is because a few demos cannot densely cover the initial object distribution and thus requires more structured generalization via warping. [[x1Cz Q3](https://openreview.net/forum?id=FqDmvMZish&noteId=6XcaHloU3D)]
4. **Tether with alternative waypoint extraction mechanisms.** In our method, we use gripper state change and finger position to extract waypoints for correspondence and warping, which is commonly done in prior work. To show that this choice of waypoint extraction is orthogonal to the core method, we run a “Pouring from Cup to Bowl” task: our original mechanism is incompatible for this task as the gripper is holding the cup, so we instead predict waypoints with VLM queries. Here, our policy achieves 90% success, thus demonstrating that Tether can readily adopt alternative waypoint extraction mechanisms to expand task and object diversity. [[x1Cz Q5](https://openreview.net/forum?id=FqDmvMZish&noteId=tuY1CUqW09)]
5. **Tether for new articulated objects.** Our original experiments demonstrated semantic generalization to unseen fruits and containers. To additionally show that our method is robust for cross-object generalization with articulated objects, we run a “Lifting Coffee Machine Handle” task using 10 demos performed on one machine and evaluated on a second unseen machine. Our policy achieves 80% success, thus demonstrating semantic generalization for a more complex task involving delicate articulation. [[x1Cz Q6](https://openreview.net/forum?id=FqDmvMZish&noteId=tuY1CUqW09)]
6. **Reactivity and recovery.** We employ a non-parametric policy design that executes action plans open-loop. While a reviewer suggests that this may limit task complexity and recoveries, we point out that within the low data regime, every prior approach for few-shot imitation requires similarly strong inductive biases. Like many past works, Tether operates on the initial state distribution to avoid compounding error over timesteps. Still, it can handle diverse and difficult tasks, comparable to or more difficult than representative prior work, as well as execute long-horizon plans and recover from mistakes during play. [[x1Cz Q1](https://openreview.net/forum?id=FqDmvMZish&noteId=wd0C5yLbTL), [x1Cz Q7](https://openreview.net/forum?id=FqDmvMZish&noteId=tuY1CUqW09)]

---

> ### Author Response · Authors · 2025-12-03
>
> Lastly, we summarize additional key discussions and experiments below:
> 1. **Scalability.** As reviewers request details on Tether’s inference speed and scalability, we point out that Tether is purposefully designed solely for the very-low data regime, with built-in structural priors that are unnecessary at higher data regimes. Our policy computes the full trajectory in ~1.6 seconds (for 10 demos), which enables our high throughput during play: 1 success every 86 seconds, including trajectory execution time. [[ZDhN Q2](https://openreview.net/forum?id=FqDmvMZish&noteId=Rwp9tu0MpW), [x1Cz Q9](https://openreview.net/forum?id=FqDmvMZish&noteId=EhpXO13hkx)]
> 2. **Cyclic Task Structure.** Our play procedure is performed on a set of tasks that satisfy a “cyclic” structure. As a reviewer requested details on this design, we clarify that the tasks are selected such that there is at least one valid task for any potential resulting environment state, thus enabling autonomous play without human resets. This is not an overly restrictive criteria, since the majority of tasks are invertible; for most tasks that are teleoperated or evaluated in real, we can construct our task structure by replacing human resets with the equivalent robot task. [[x1Cz Q2](https://openreview.net/forum?id=FqDmvMZish&noteId=wd0C5yLbTL)]
> 3. **Correspondence accuracy.** Tether leverages semantic correspondences between a demo and the current scene to compute its trajectory warping. While reviewers point out that our policy is dependent on the accuracy of these correspondences, we empirically observe that as Tether inherits recent advances in semantic correspondence based on vision foundation models, it is robust to textureless surfaces, deformable regions, lighting variations, and partial occlusions. Additionally, Tether handles noisy correspondences by filtering out failed backprojections. [[ZDhN Q1](https://openreview.net/forum?id=FqDmvMZish&noteId=Rwp9tu0MpW), [CA2Q Q2](https://openreview.net/forum?id=FqDmvMZish&noteId=ESBJul5hAt)]
> 4. **Tether in new environments.** Our original experiments are performed in a cluttered household-like scene with many objects. To demonstrate Tether’s effectiveness in additional settings, we run “Pineapple to Bowl” in a new mock kitchen setup and office setup. Our policy achieves 100% and 90% success respectively, thus validating that our method works well across geometrically and structurally diverse environments. [[QvvQ Q2](https://openreview.net/forum?id=FqDmvMZish&noteId=OzYeT5u5pC)]
> 5. **Tether in additional multi-object scenes.** Our original experiments show that Tether is performant in multi-object settings (eg, with a fruit and container). To additionally validate Tether’s ability to disambiguate more semantically similar objects, we run the “Fruit to Bowl” task for a pineapple, strawberry, and apple, with all three fruits on the table in all trials. Our policy achieves 100%, 80%, and 100% success respectively, which confirms that it accurately distinguishes between similar objects in cluttered multi-object scenes. [[QvvQ re-Q6](https://openreview.net/forum?id=FqDmvMZish&noteId=3sWLSbB1nC)]
>
> We are revising our manuscript with these improvements and clarifications. We believe that our experiments are comprehensive, and that our paper overall makes important contributions to the long-standing data scarcity problem in robot learning.
>
> Best,
> Authors
>
> [1] Di Palo et al. Keypoint Action Tokens Enable In-Context Imitation Learning in Robotics
> [2] Fang et al. KALM: Keypoint Abstraction Using Large Models for Object-Relative Imitation Learning.

---

### Meta-Review · Area_Chair_Ezek · 2026-01-07

**Summary:**

The paper introduced an automatic data collection pipeline that bootstraps from a small amount of existing robot demonstrations. The key idea is to leverage semantic correspondences to associate objects of interests and then warp the trajectories into the new scene. One nice interesting design is its cyclic task structure, which allows the initial state distribution of tasks to grow as the object configurations drift away from the beginning of play. With the automatically collected data, the authors show that they can learn a much better policy. Of course, the method is far from perfect and is limited in various ways: for example, it cannot handle occlusion; it does not consider the geometry of the scene (eg, assuming no collision from the environment while executing the trajectory); it is restricted to open-loop setup; and its performance is bounded by the perception module, etc. The obvious pros and cons have thus resulted in diverging reviews (ie, 2, 4, 4->6, 8, 8). During the rebuttal, the authors provided extensive experiments. While the inherent limitation of the approach is still there, the additional results did shed light on the capabilities of the proposed framework. After extensive discussion, the ACs find the paper still have its merit and the pros outweigh cons. The ACs thus decide to accept the paper. The AC urges the authors to incorporate the feedbacks from the reviewers into their final version.

**Reviewer Concerns:**

See above.

**Reviewer Scores:**

See above.

---

### Decision · Program_Chairs · 2026-01-26

Accept (Poster)